# A basic macroeconomic agent-based model for analyzing monetary regime shifts

**Florian Peters**[1]*, **Doris Neuberger**[1], **Oliver Reinhardt**[2], **Adelinde Uhrmacher**[2]

**1** Department of Economics, Faculty of Economic and Social Sciences, University of Rostock, Rostock, Germany, **2** Visual and Analytic Computing, Faculty of Computer Science and Electrical Engineering, University of Rostock, Rostock, Germany

* florian.peters@uni-rostock.de

**Data Availability Statement:** Data is available on the following repository (folder "Simulation replications"): https://github.com/fhaegner/Mak-h-ro.

## Abstract

In macroeconomics, an emerging discussion of alternative monetary systems addresses the dimensions of systemic risk in advanced financial systems. Monetary regime changes with the aim of achieving a more sustainable financial system have already been discussed in several European parliaments and were the subject of a referendum in Switzerland. However, their effectiveness and efficacy concerning macro-financial stability are not well-known. This paper defines the economic requirements for modeling the current monetary system and introduces the corresponding macroeconomic agent-based model (MABM) in a continuous-time stochastic agent-based simulation environment with a provenance model. This MABM aims to present a starting point for exploring and analyzing monetary reforms. In this context, the monetary system affects the lending potential of banks and might impact the dynamics of financial crises. MABMs are predestined to replicate emergent financial crisis dynamics, analyze institutional changes within a financial system, and thus measure macro-financial stability. The used simulation environment makes the model more accessible and facilitates exploring the impact of different hypotheses and mechanisms in a less complex way. Moreover, the model replicates a wide range of stylized economic facts, which validates it as an analysis tool to implement and compare monetary regime shifts.

## 1 Introduction

The financial system is a complex system [1] that is inherently prone to fragility. During the last decades it has been subject to various crises [2]. Notably, the crisis of 2007/2008 triggered a debate about possible alternative monetary system approaches and how to improve the stability of the financial system towards endogenous and exogenous shocks.

The monetary system is a central constituent of the financial system, and embraces accounting measures concerning banks' credit creation potential and the institutional organization of financing economic activity. It is the underlying "software" of the financial system that enables payments, savings, insurance, and financing. The main system ingredient relates to the fractional reserve system which defines the interplay of the Central Bank, banks, and the

**Funding:** The funder is the DFG - Deutsche Forschungsgemeinschaft with the project number 258560741 and project title: Modeling and simulation methods for linked lives in demography. Prof. Dr. Adelinde Uhrmacher (AU) is the applicant and project leader, and Oliver Reinhardt (OR) is a team member of the project.

**Competing interests:** The authors have declared that no competing interests exist.

real economy to ensure the financial scheme or credit creation structures. A more detailed description is provided in Section 3.1.

The current debate on monetary reform is controversial (e.g., Deutsche Bundesbank [3] vs. Huber [4]). The main issue of dissent is whether the current system is inefficient and ineffective in terms of macro-financial stability compared with available alternatives (e.g., optimal money supply, effective financial intermediation, shadow banking). There are two contrary reorganization approaches to the current monetary system that both claim to lead to a less fragile financial system: First, the sovereign money system that **centralizes** money creation towards sovereign control via a strict separation of money and credit [4]. A more detailed overview and recent political developments are summarized by KPMG [5]. And second, the free banking system that **decentralizes** money creation with little or no sovereign interference [6].

To lay the foundation for analyzing such "what-if" scenarios and compare different monetary systems concerning macro-financial stability features, this paper provides a baseline simulation model. It focuses on modeling the current monetary system as a benchmark since the simulation output can be calibrated and validated against empirical evidence. The implementation of alternative monetary systems and discussion of associated economic implications is the task of future research within this framework.

To measure (in)stability conditions, the corresponding analysis has to incorporate the emergence of systemic risk. According to Borio [7], Bank of England [8], and Galati and Moessner [9], systemic risk is classified as time-varying and cross-sectional risk. Time-varying risk belongs to the pro-cyclical behavior of the financial sector, caused by solvency, liquidity, and contagion risk [10], which drives the financial cycle [11]. Cross-sectional risk is related to the structural composition of the financial sector, such as the distribution of risk within the financial network, co-varying risks across balance sheets, systemically important financial institutions and opacity of complex network interdependencies [10].

Two simulation approaches appear to be eligible for such a macroeconomic analysis: Dynamic Stochastic General Equilibrium (DSGE) models and Macroeconomic Agent-Based Models (MABM). Since we are interested in examining feedback effects between economic sectors arising from financial instability dynamics, DSGE models appear to be more limited than MABM. According to Alexandre and Lima [12], "a key element to grasp the occurrence of financial crashes is an understanding of nonlinear feedbacks among financial and real variables". Incorporating nonlinearities is limited within DSGE models [12] but a fundamental feature of agent-based models [13]. Furthermore, Bookstaber [14] argues that the representative agent used in DSGE models, which involves optimizing behavior and full rationality assumptions, is not appropriate for analyzing crisis dynamics, such as excessive leverage, funding fragility, liquidity limits, coordination failures, market freezes, and bankruptcies cascades. This is because such assumptions are based on historical relationships, which are not valid during instability phases [12]. In contrast, the aggregate dynamic of MABMs is based on the emergent property via the interaction among adaptive heterogeneous agents [15–17]. Further building blocks of agent-based models, and a detailed comparison between DSGE and MABM can be seen in [13, 18–21]. Lastly, socio-economic systems are inherently non-stationary [13], stochastic, non-ergodic, and structurally evolve over time [22], making MABMs a natural fit.

Macroeconomic models with focus on analyzing monetary reform are scarce and do not clarify the main controversial issues: Benes and Kumhof [23] use a DSGE model to analyze a full reserve monetary regime (100% reserve system) and conclude an elimination of bank runs, better control of business cycles and reduction in private and public debt. However, they do not differentiate between the effects of macroprudential policy and monetary reform, which could distort the results. Furthermore, the origination of (interest bearing) reserves and

influence of money supply is unclear. Similar results can be derived from the MABM of van der Hoog [24], even though money supply and creation do not play a role in the analysis, which should be an essential component. The author simulates a full equity finance intermediation involving a static aggregate money supply. Both models refer to a full reserve system (100% money [25]) with synchronization rather than separation of money and credit (narrow banking). However, the split circuit between money and credit is seen as a source of an unstable monetary system [4]. Furthermore, a monetary reform is often misunderstood with a banking reform and should be differentiated in the model. The current debate comprises the sovereign money system, which strictly differentiates between central bank money and private bank/debt money (liquidity money). In addition, the discussion is related to a debt-financed approach [4, 26] where banks act as pure financial intermediaries, which includes maturity transformation and corresponding liquidity risks, and not as equity fund managers without refinancing risks [24]. Such contextual intricacies and the corresponding model requirements (see Section 3.1 for more details) need to be implemented and discussed when analyzing monetary reforms.

Hence, the contribution of this MABM to the literature is twofold, comprising methodological and economic aspects: The methodological contribution consists of a specified MABM that is implemented in a way to allow for successive extensions and easy exchange of central model components to validate the effectiveness and efficacy of a monetary reform with respect to macro-financial stability measures. Despite the inherent complexity of the financial system, it is expedient to easily access and discuss the model, not only to work with it, but also to assess its quality. Therefore, we use a compact and expressive domain-specific language that facilitates more succinct and accessible model specifications rather than a realization in a general-purpose language [27], and include a provenance model of our MABM (see Section 2.2 and Fig 3).

The economic contribution of this MABM refers to the definition of the economic requirements (Section 3.1) for modeling the current monetary system and the corresponding basic model that captures the main systemic risk features to implement and compare alternative monetary regimes. To our knowledge, no MABM (nor DSGE model) exists to analyze a sovereign money or free banking system. However, we find a few agent-based models that incorporate parts of the current monetary system to analyze the impact of policy variations [24, 28–30]. To model the monetary system, we define six economic requirements, whereby the main contribution of this MABM is emphasized with the following four requirements: First, the current fractional reserve system and the associated endogenous reserve demand, which is based on the interaction between real and financial sector. Second, the macroprudential framework and its impact on macro-financial measures to extract the effect of a monetary reform. Both requirements are not adequately or only partially implemented in corresponding MABMs [24, 28, 30]. The same holds true for modeling the financial cycle [29, 30] and endogenous money creation [24, 30]. One MABM [29] mainly fulfills the defined model implementation but is not available or replicable, which underlines the methodological contribution mentioned above. Further details and related literature are discussed in Section 6.

The MABM at hand is subject to a validation which shows that the presented model is a plausible representation of the current monetary system. Therefore, this MABM is a starting point for exploring and analyzing monetary reforms (see Section 5 for more details).

The paper is structured as follows. In the following Section materials & methods are described. The requirements for the model and corresponding model specifications are presented in Section 3. Afterward, the calibration process is described in Section 4 and simulation validation results are shown in Section 5. Finally, Section 6 compares the contribution of the presented model to related literature.

## 2 Materials and methods

### 2.1 The modeling language for linked lives

To realize the model, we make use of the Modeling Language for Linked Lives (ML3) [31]. ML3 has been originally designed for agent-based demographic models, in which live-courses of interacting individuals are simulated. However, its application is not limited to demography, and the economic processes could be easily mapped to its modeling concepts.

At the core of ML3 is the concept of stochastic rules, which are used to model the behavior of agents. Each rule is associated with one type of agents (e.g., households, firms, banks, . . .) and consists of the following three parts:

First, the **guard** is a condition that allows to specify to which agents of that type the rule applies.

Second, the **rate** determines when the rule is executed. It defines the parameter of an exponential distribution from which the waiting time is drawn. In some rules this may be replaced by a fixed waiting time, resulting in periodic events, or no waiting time at all, in which case the rule is applied instantly.

Third, the **effect** determines what happens when the rule is executed. As an example, we may want to model that firms produce some goods at a certain rate. For simplicity's sake, we assume that every agent of type firm produces goods, hence the guard condition should always be fulfilled, i.e., `true` (see Fig 1, line 2). The production rate may vary between the different agents of type firm. It is saved as an attribute `production rate` of the firm-agent. The keyword `ego` gives access to the specific agent a rule is applied to, and the production rate for this agent is then `ego.production rate` (see line 3). The effect of this rule should be that the stock of inventories increases by 1 (see line 4).

During the simulation, a waiting time is drawn for each rule of every agent. The rule and agent with the lowest waiting time are selected and the rule is executed for that agent. The simulation time is then advanced by that waiting time. The resulting underlying mathematical processes, i.e., the semantics of ML3, are formally defined as Generalized Semi-Markov Processes in [31]. The model description in Section 3.3.1 and 3.3.3 provides further exemplary applications of the rule concept.

ML3's rule-based modeling approach completely separates model logic (i.e., the behavior of agents) and simulation logic (i.e., the execution of that behavior). The former is implemented by the modeler using the modeling language of ML3. And the latter has been implemented independently of a specific model by the developers of the modeling language. This separation of model and simulator improves accessibility both for the modeler and a later reader of the model. They do not need to be concerned with implementation details of the simulator, but only with specifying the model within the modeling language.

The stochastic rules allow for viewing the different kinds of behavior an agent can exhibit (e.g., for a firm: production, hiring, investment, and others) independently of each other. As

```
1  Firm
2  | true
3  @ ego.production_rate
4  -> ego.inventory := ego.inventory + 1
```

**Fig 1. An example of a rule, modeling the production of a good at a certain rate.**

the simulation does not advance in time steps, no priority among or ordering the various behaviors of different agents needs to be specified. Each rule fires with its own rate sampled from an exponential distribution. This allows us to separate the model into different components, each consisting of only a few rules, describing a part of the model, and to combine components step by step to yield the complete model.

## 2.2 Provenance

For documenting the simulation model and its foundation in economic theory we complement a textual description of the model with a provenance model [32, 33]. While the former focuses on the mechanics of the model, the latter captures information about the context of the model, and the process of its creation. It relates the components of the model to economic theories they are based upon, and shows the central steps in the model's creation. This adds crucial information for assessing the simulation model itself, as well as the results it generates, by making its foundation, and therefore its qualities and limitations, more transparent.

The provenance model is based on the PROV standard [34]. Thereby, the provenance is modeled as a directed acyclic graph containing two types of nodes (see Fig 3). **Entities** (shown as circles) represent the different artifacts involved in the modeling process, both as inputs (e.g., data, earlier models, economic theory), and as products (e.g., model components, the complete model). **Activities** (shown as squares) represent the processes that link them (e.g., modeling, composing components, performing simulation experiments). The edges of the graph relate the entities and activities explicitly and visually, showing which entities were *used by* a given activity, or which entities were *generated by* it.

## 3 Model outline

As described in the introduction, this paper serves as a methodological amendment to macroeconomic agent-based models (MABMs) to analyze alternative monetary system regimes concerning macro-financial stability. The requirements for such a (baseline) model are specified in the following paragraphs.

## 3.1 Model requirements

To measure and compare financial and macroeconomic stability features at the aggregate level of an economy (macro-financial stability), the interrelation between the real and financial sectors is crucial to capture credit demand and supply and the associated feedback effects.

Hence, a **business cycle** *(requirement 1)* is needed which defines the real sector and comprises the interaction of households (consumption behavior) and firms (investment behavior) to take credit demand into account. Furthermore, the business cycle allows to grasp the economy's performance (e.g., GDP—Gross Domestic Product, employment, inflation, firm bankruptcies) and serves as a foundation to measure the impact of a reorganized credit supply.

The **financial cycle** *(requirement 2)* defines the credit cycle [35–38], specifically banks' credit supply in interaction with the real sector. As suggested by Borio [11], the financial cycle needs to be modeled in the way that "[...] the financial system does not just allocate, but also generates, purchasing power, and has very much a life of its own [...]", which is in line with the non-neutrality of money assumption [39] and congruent to the methodological approach of MABMs as described in the introduction. To analyze systemic risk that might arise from overshooting credit cycles, as claimed by proponents of sovereign money [4], the overinvestment hypothesis needs to be considered. The investment behavior of firms can be characterized as a state of under- or overinvestment due to asymmetric information and the circumstance that firm managers tend to maximize their utility rather than the firm value.

In addition to firms' behavior, banks contribute to overshooting credit behavior by profiting from supplying more money than absorbed by the production capacity and causing unsustainable credit growth [11]. Here, a clear indication of credit creation potential is needed which banks strive to exploit and expand. Furthermore, the distinction between productive and unproductive credits is expedient to identify macro-financial stability effects of different monetary regimes. Unproductive credits might be induced by rolling over of existing debt, leading to unnecessary prolongation of firm existence with high default risk [24, 40].

Next to systemic risk sources described in *requirements 1 and 2*, the main focus is steered to the organization of money creation, which is covered by the following *requirements*.

The **fractional reserve system** *(requirement 3)* defines the structure of the current monetary system which is a two-tier split-circuit system in accordance with the circuit theory [4]. A two-tier structure consists of a Central Bank and a private banking sector. Split-circuit describes the dichotomy of banking money (public circuit) and central bank money (interbank circuit) [4]. Banks need only a fractional amount of reserves (central bank money) to refinance credit receivables and process payments that bear liquidity risks (see *requirement 4*). This distinction between money and credit is essential to compare alternative systems. A sovereign monetary system can be defined as a two-tier single-circuit money system (central bank money as the only legal tender) and free banking as a one-tier single-circuit money system (only private bank money circulation without Central Bank intervention). Hence, the implementation requires an endogenous payment scheme and credit network.

**Endogenous money creation** *(requirement 4)* describes the money issuance process between Central Bank, banks, and the real sector according to the prevailing opinion [3, 41]. Within the current system, banks' credit supply to firms plays a key role, which involves creation of deposits (maturity transformation). Banks are exposed to credit default risks, interest-change risks (arbitrage between long-term credit and short-term deposit interests) and liquidity risks. Proponents of alternative monetary systems claim that liquidity risk might be mitigated [4] because the co-evolution of the re-organized credit and interbank market is more stable than the current system. This would prevent liquidity freezes and related knock-on or spillover effects to the real sector.

**Macroprudential regulation** *(requirement 5)*, according to the Basel III framework [42], is a mechanism to attenuate systemic risks originated by banks described in the previous model *requirements*. The regulation framework includes banks' capital and liquidity requirements, which might reduce unproductive credit creation (less overshooting credit cycle) and dampen crisis dynamics (less severe downturns and bank bailouts). Since alternative monetary regimes address the same systemic risk, this mechanism is crucial to compare the de-risk effectiveness.

The **basic monetary transmission mechanism** *(requirement 6)* indicates the impact of monetary policy by the Central Bank on general economic conditions and, in particular, the price level. This transmission is influenced by the monetary system via transmission channels and identifies feedback effects between the real and financial sectors. According to the single monetary policy in the Euro area [43], interest rate changes by the Central Bank affect firms' investment decisions via funding costs (capital cost and income effect) and correspondingly credit demand (interest rate channel). Interest rate changes also affect credit supply (credit channel). For example, an increase in interest rates negatively affects the financing conditions of firms and banks charge a risk premium which might lead to credit selection and rationing (bank lending channel of the credit channel). Furthermore, interest rate changes affect the net worth of firms and their expected cash flow, which in turn affects firms' collateral and the ability to borrow (balance sheet channel of the credit channel). Such fluctuations in the creditworthiness of borrowers might lead to an acceleration of credit demand, known as financial accelerator [44]. This pro-cyclical behavior also affects the risk-taking behavior of banks and

borrowers. The expectation of a sustained increase in collateral values in boom phases combined with low-interest rates leads to higher risk-taking. Banks might soften credit standards, which can boost excessive lending (risk-taking channel). All channels are interdependent and lead to aggregate demand and price changes, thereby affecting the business cycle (*requirement 1*) and credit cycle (*requirement 2*). For simplicity, the expectation channel and indirect effects via financial portfolio decisions, and exchange rate changes are excluded in the basic model.

### 3.2 Model overview

The implementation of the defined model requirements in ML3, described in the previous Section, is gradually composed of submodels, as shown in Fig 3 and described in the following Sections.

The agent-based model structure is based on five different agents types (households, firms, banks, Central Bank, and government) which interact within and between the real and financial sector, as shown in Fig 2.

The real sector is composed of the following market segments:

- A labor and goods supply market where households selling their labor (employees) to firms to produce homogeneous goods in exchange for wages to consume (**labor** and **goods supply** specification in Fig 3). Wages are households' financial wealth used for payments (deposits) and are entirely consumed.

- A consumption market where households interact with firms to generate consumption demand (**goods demand** specification in Fig 3).

- An investment market where firms interact with each other to generate investment demand (**investment market** submodel in Fig 3).

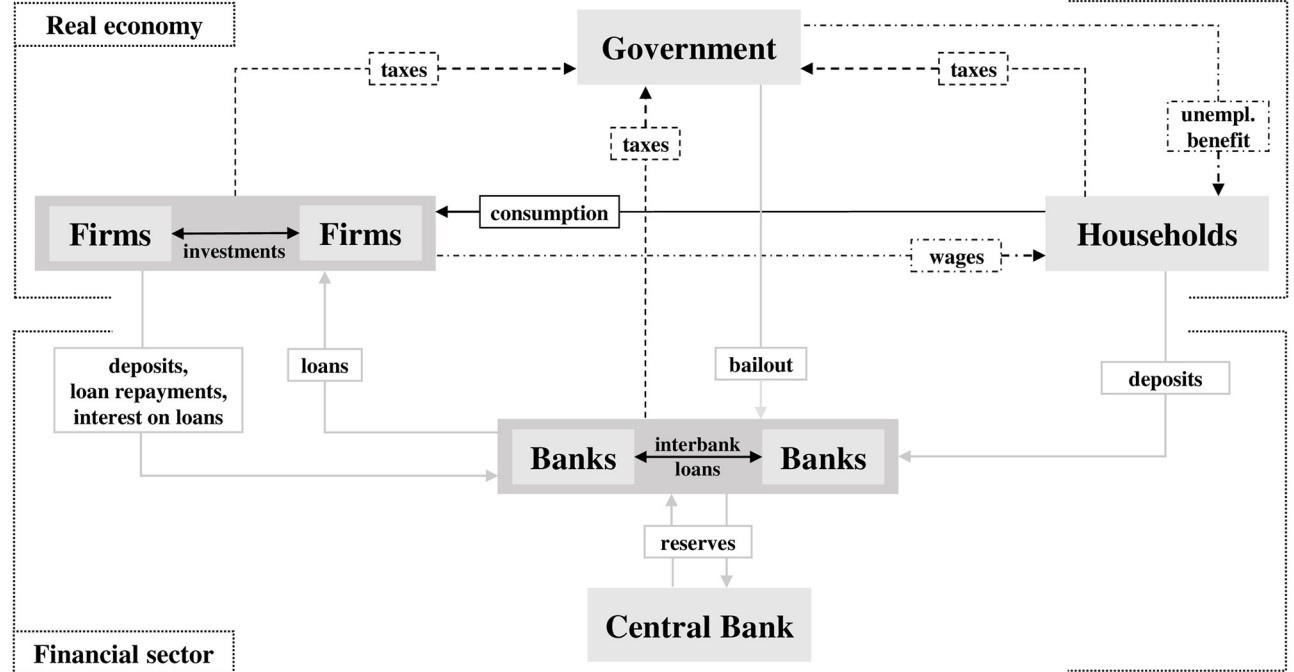

**Fig 2. Stylized stock-flow structure following Caiani et al. [45].** Arrows point from paying to receiving agent.

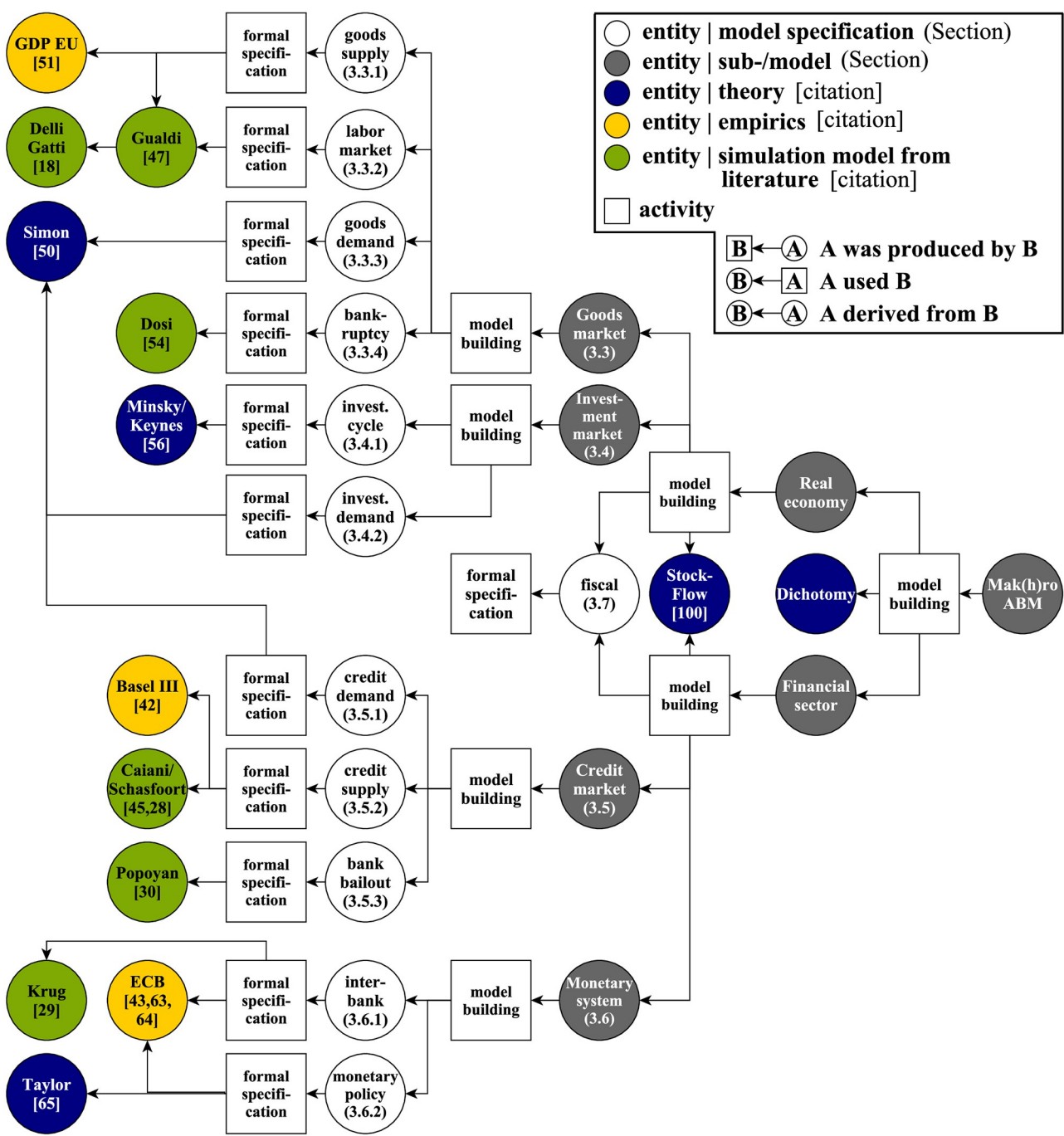

**Fig 3. Provenance model. Note:** The Figure shows the provenance of the composed macroeconomic agent-based model "Mak(h)ro_0" including entities from theory, empirics and simulation models from literature. Section references are labeled in parentheses and citations in brackets. For a short description of the entities and activities in this Figure, see S1 Table.

Whereas the financial sector is composed of the following market segments:

- A credit market where firms interact with banks to finance investments, production/supply, and outstanding debt (**credit market** submodel in Fig 3). According to the endogenous monetary theory [3, 41], banks create loans and deposits for firms and indirect deposits for households via wages.

- An interbank market where banks interact with each other and the Central Bank to guarantee financial transactions and monetary policy transmission (**interbank** and **monetary policy** specification in Fig 3). All transactions are account-based and flow between balance sheets of all agents, and are settled by banks in combination with the interbank market. A more detailed balance sheet overview of all agent types can be found in S2 Table.

- A deposit market where households and firms hold deposits at bank accounts, which is included within the interbank market.

The public sector is set by the Central Bank and government. The Central Bank controls the flow of reserves via open market operations and influences the interest rate level for bank loans. The government collects taxes and redistributes them to households and banks via unemployment benefits and bank bailouts.

**3.2.1 Design concept and limitations.** The model represents economic activities between heterogeneous agents based on heuristic behavior rules to emphasize the complexity of the allocation process in the real world and to simulate emergent financial crisis dynamics. An agent's behavior rules do not change over time and have no learning capabilities. Time is measured in periods, thus a transition rate of 0.5 would represent two waiting periods on average. Besides accounting measures (e.g., wages, profits, loan repayments, etc.), agent behavior rules are not bound to sequence or discrete events but rather occur concurrently in continuous time, as described in Section 2.1. Agent expectations are based on simple averages of past events (i.e., cash flow expectation in Section 3.5.2), and the matching process between agents within implemented market structures follows a weighted random selection (see lines 11 and 12 in Fig 5).

Concerning limitations on the modeled economic structure, the following simplifying assumptions are made: The agent population, labor productivity, and wages are fixed over the entire simulation run. Thus, the model cannot capture long-term growth. Firm investments function as demand drivers but are not implemented in firms' production processes. The model does not incorporate a capital or bond market, and households do not own firms to keep the necessary structures simple. Hence, it is assumed that government expenditures are not restricted (e.g., for bank bailouts). This circumstance makes the model not stock-flow consistent, according to Godley and Lavoie [46]. Other financial sources are bank loans and deposits. Cash money/banknote holding is not assumed. Finally, the model does not incorporate shadow banking.

In the following sections, we will describe and discuss the different submodels, the entities, and activities that contributed to their development following the provenance model as depicted in Fig 3.

## 3.3 Goods market

The real economy consists of two submodels (goods and investment market) which simulate the business cycle (*requirement 1* in Section 3.1). The goods market builds on four central mechanisms for which separate model specifications have been developed, as shown in Fig 3.

**3.3.1 Supply mechanism of firms.** A fundamental component in the real sector is the behavior of firms and their adaptation to demand impulses. As apparent in Fig 2, demand is based on consumption by households (Section 3.3.3) and investments by other firms (Section 3.4). The supply mechanism models the update rule for pricing and production based on a heuristic behavioral rule, according to Gualdi et al. [47]. Its implementation in ML3 is shown in Fig 4. The production is based on a homogeneous good used for consumption and

```
1   Firm
2   | true
3   @ ego.strategy_change_rate()
4   -> ego.set_strategy()
```

**Fig 4. Update rule of firms for prices and production quantity in ML3. Note:** The Figure shows the supply mechanism rule, modeling firm production behavior as shown in Eq 1 at the transition rate defined in Eq 2.

investment, which is common in macroeconomic agent-based simulation (see an overview by Dawid and Delli Gatti [48]).

To make it simple, it is assumed that labor productivity $\alpha$ is fixed over time and capital is not a production input. The production capacity $Y_i$ of firm $i = 1...N_F$ is a product of the hired amount of labor $\Lambda_i$ and labor productivity.

To adapt to demand impulses, firms determine their labor demand $\Lambda_i^T$ via setting a production target $Y_i^T$. At the same time, the firm needs to find a sensible price $p_i$ for their good. Both together form the firm's strategy (ego.set strategy()), which is updated as follows:

$$Y_i \leq D_i \text{ and } p_i \geq \bar{p} \text{ and } u > u_{nat} \quad \Rightarrow \quad Y_i^{T\prime} = Y_i \cdot [1 + \gamma_y \cdot \xi]$$

$$Y_i \leq D_i \text{ and } p_i < \bar{p} \text{ and } u > u_{nat} \quad \Rightarrow \quad p_i' = p_i \cdot [1 + \gamma_p \cdot \xi]$$

$$Y_i \leq D_i \text{ and } u < u_{nat} \quad \Rightarrow \quad p_i' = p_i \cdot [1 + \gamma_p \cdot \xi] \tag{1}$$

$$Y_i > D_i \text{ and } p_i \leq \bar{p} \quad \Rightarrow \quad Y_i^{T\prime} = Y_i \cdot [1 - \gamma_y \cdot \xi]$$

$$Y_i > D_i \text{ and } p_i > \bar{p} \quad \Rightarrow \quad p_i' = p_i \cdot [1 - \gamma_p \cdot \xi]$$

where $D_i$ represents the demand for the firm's goods since the last strategy change and $\bar{p}$ the current average price of all firms. To control the price path, and since wages and productivity are fixed, the average price is calibrated at a fixed value of 1.5. Hence, the price fluctuates around the average level. The parameters $\gamma_y$ and $\gamma_p$ define the sensitivity to production and price adaptions, respectively. $\xi$ is a random variable $U[0, 1]$ to implement uncertainty.

An extension to Gualdi et al. [47] is the full employment state where firms only increase their price. This state is reached when the aggregated unemployment rate $u$ is lower than the Non-Accelerating Inflation Rate of Unemployment (NAIRU), which is assumed to be 5% and in line with advanced economies before the financial crisis 2007/2008 (OECD statistics on structural unemployment). The behavior described in Eq 1 is subject to the profit orientation of firms. Firms tend to get more market share or exploit arbitrage opportunities in a monopolistic price setting. The cost-covering price for each firm is based on wage and borrowing cost per unit and defines the minimum price.

The transition rate $rate_i^{prod}$ (ego.strategy change rate()) is dependent on the firm's stock of inventories variable $N_i$ concerning their stock target of inventories $N_i^T$, which is a parameter and assumed to be twice as high as the production capacity:

$$rate_i^{prod} = \frac{N_i}{N_i^T} \text{ or } 1 + \left(1 - \frac{N_i}{N_i^T}\right) \text{ if } rate_{prod} < 1 \tag{2}$$

The higher the delta between current inventories and inventory target, the higher the strategy transition rate and vice versa. This has the effect that firms hold a buffer of real inventories against unexpected demand swings [49] and avoid supply constraints [45]. In conjunction

with Eq 1, the effect is that prices and production capacity constantly increase in boom phases and the other way around in recessions. Combined with the following model specifications, this effect leads to a typical business cycle behavior (*requirement 1*).

**3.3.2 Labor market.** To reach the firms' targeted production capacity (Eq 1), each firm needs to hire or fire labor. The labor market operates on a simple hire-fire principle following Gualdi et al. [47], as shown in Fig 3. The rates for hiring and firing are dependent on the mismatch of labor demand $\Lambda_i^T$ to current labor $\Lambda_i$. The greater this mismatch, the higher the rate for hiring (or firing) employees:

$$\text{hire rate} : \left(\frac{\Lambda_i^T}{\Lambda_i}\right)^{\zeta_h} \text{ if } \Lambda_i^T > \Lambda_i$$

$$\text{fire rate} : \left(\frac{\Lambda_i}{\Lambda_i^T}\right)^{\zeta_f} \text{ if } \Lambda_i^T < \Lambda_i$$

(3)

Thereby, $\zeta_h$ and $\zeta_f$ are sensitivity parameters defined for the hire and fire process respectively, and set via calibration. The effect of the rules is hiring or firing one unit of labor.

**3.3.3 Demand mechanism of households.** The nexus between supply and demand is coordinated by the price for a homogeneous good. The consumption rate (line 3 of Fig 5) is dependent on a household's financial wealth, which consists of wages or unemployment benefits as shown in Fig 2.

The rate indicates that households aim to fully consume their income during each period unless constrained by supply (`firm.size() > 0`) or prices (`ego.wealth > firm.price`), shown in line 4 of Fig 5. The selection process is modeled as a weighted random selection (lines 11, 12). The function `weightedRandom` returns a random element of the set whereby the probability of selecting a particular firm $i$ with price $p_i$ is proportional to its weight. Only `firms` are considered (`filter()`) within the entire agent population of firms (`Firm.all`) with a good left in their stock of inventories (`alter.inventory > 0`) (line 10). After a successful match, the transaction is completed in lines 5 to 8. The `demand` variable of a firm is reset after each strategy update, described in Section 3.3.1.

The stochastic nature of the matching protocol serves to model incomplete price information of the households according to bounded rationality [50]. Households prefer to select a

```
1   Household
2   | true
3   @ ego.wealth
4   -> if (firm.size() > 0 & ego.wealth > firm.price) then
5         ego.wealth  -= ?firm.price
6         firm.liquidity += ?firm.price
7         firm.inventory -= 1
8         firm.demand += 1
9      end
10  where firms := Firm.all.filter(alter.inventory > 0)
11        firm  := firms.weightedRandom(
12        Mercury.average_Price() / alter.price)
```

**Fig 5. Household consumption rule in ML3. Note:** The Figure shows the demand rule, modeling household consumption or expenditure of financial wealth in exchange for homogeneous goods from firms. "Mercury" is an auxiliary agent, which otherwise has no interactions with other agents.

supplier with a price below average but will usually not be able to find the one with the lowest price.

The share of consumption to nominal GDP is calibrated to culminate at around 50% according to the Euro area in recent years [51] (see Fig 3). This also includes government expenditures (Section 3.7). The remaining 50% is related to demand generated by firm investments (Section 3.4).

**3.3.4 Bankruptcy of firms.** The bankruptcy mechanism belongs to an accounting process and defines when and how a firm goes bankrupt. The rule is not dynamic and is only activated once a period when the debt ratio (Eq 10) exceeds the insolvency hurdle. The insolvency hurdle is assumed at a debt ratio of 200%, which is calibrated in combination with the Minsky base cycle described in Section 3.5.1.

The rule's effect is that all credit lines are terminated, thereby burdening banks' equity (see Section 3.5.3). Employees are fired up to a calibrated minimum size of five employees. Hence, firms are assumed not to be reborn and are, on average smaller than their non-bankrupt counterparts, which is in line with Caves [52] and Bartelsman et al. [53], found in Dosi et al. [54] (see Fig 3).

## 3.4 Investment market

Next to household consumption (Section 3.3.3), the second demand pillar is the investment demand and the corresponding market implementation. The investment market is modeled endogenously in a non-growth environment, according to Lengnick [55] and Popoyan [30], to keep the model structure as simple as possible. There are no capital firms, no technological changes, and investments are not accumulated on the balance sheet.

**3.4.1 Investment cycle.** To incorporate over- and underinvestment and to set the basis for an overshooting credit cycle (*requirement 2* in Section 3.1), the investment process is based on the stylized theoretical approach by Keynes [56] (see Fig 3), where firms indirectly generate their future profits with current investments in a macroeconomic perspective. The higher the aggregate investments of firms, the higher the average demand of all (other) firms in upcoming periods. From a microeconomic perspective, the demand generated by a single firm triggers other firms to increase investments, which induces demand for the firm that initially set investment impulses.

The update rule of firms in ML3 to set their targeted investment amount $inv_i^T$ follows the same structure as Fig 4 with the following strategy:

$$
\begin{aligned}
ret_i > 0 \text{ and } inv_i \leq lim_i &\Rightarrow inv_i^T = inv_i \cdot [1 + \gamma_{inv} \cdot \xi] \\
ret_i > 0 \text{ and } inv_i > lim_i &\Rightarrow inv_i^T = lim_i \\
ret_i < 0 &\Rightarrow inv_i^T = lim_i
\end{aligned}
\tag{4}
$$

where $ret_i$ represents the investment return of previous investment cycles ($t \rightarrow n$). The calibration parameter $\gamma_{inv}$ regulates the sensitivity of investment amount changes, and $\xi$ is a random variable $U[0, 1]$. The liquidity limit $lim_i$ models the firm's creditworthiness on the average credit market interest rate and expected profit or, in other words, the assumption of the maximum available loan redemption rate that the firm can afford. A more detailed description of the creditworthiness mechanism is presented below in Section 3.5.1 (Eq 7). Hence, the investment amount depends on investment return.

The investment return is the difference between investment turnover and investment costs from a micro perspective. Investment turnover includes sales from firm *i* **to** all other firms *n*, and costs are investment purchases by firm *i* **from** all other firms *n*. A positive investment

return leads to higher liquidity limit/targeted investment amount due to higher expected profit and the other way around.

As noted in Eq 4, firms can set the investment amount above the limit. This mechanism meets the need to exploit profits in boom phases, ignoring the internal creditworthiness indicator. In this case, firms can only obtain a loan from banks at an interest rate below the average interest rate on the credit market. Otherwise, the credit demand is rationed (see below in Section 3.5).

The pro-cyclicality of firm investment behavior is generated with the transition rate $rate_i^{inv}$:

$$rate_i^{inv} = \text{if } \frac{S_i}{S_i^T - int_i} \leq 1 \text{ then } \frac{1}{cap_i} \text{ else } \frac{1}{cap_i + S_i} \tag{5}$$

where $cap_i$ is the distance between the average production capacity of all firms (average production capacity with full employment across all firms) and the current production of a firm. The solvency rate $S_i$ is defined as the debt ratio (Eq 10), and $S_i^T$ is the insolvency hurdle described above in Section 3.3.4. The variable $int_i$ symbolizes interest pressure, calculated as the relation between the latest loan interest and principal payments.

The lower the distance to the average production capacity, the higher the incidence of investment activities. The highest incidence is assumed to reach a three-period cycle on average. From a macroeconomic perspective, a low distance to the average production capacity across all firms tends to close the output gap (reaching full employment), which comes along with inflationary pressure. Higher prices lead to higher investment returns and again stimulate new investments. This cascade is slightly relaxed by interest pressure $int_i$. Since this pro-cyclical rally towards full employment becomes more severe over time when more and more firms get involved, the overall interest rate level is increased by the Central Bank (see Section 3.6.2), which decreases the expected profit and investment demand (interest rate channel—*requirement 6*).

On the other hand, a high distance to the average production capacity has a contrary effect with a low investment incidence. The lowest incidence is dependent on the average production capacity and the minimum number of firms' employees (Section 3.3.4). This aligns with the Minsky cycle [57, 58], where firms behave risk-averse in bust phases and risk-seeking in boom phases.

**3.4.2 Investment demand.** The market implementation of firm investments occurs when firms purchase homogeneous goods from other firms. For simplicity, no differentiation between household consumption and investment goods is made. Investment demand impulses, or the matching of investing firms to suppliers, work in the same way as the demand mechanism of households (lines 11 and 12 in Fig 5). The purchase incidence rate depends on the investment amount described in the previous Subsection (like line 3 in Fig 5).

*Funding condition.* In line with Dosi et al. [54], it is assumed that capital markets are imperfect, and the financial structure of firms matters (e.g., [59–61]). According to Myers [62], internal or external borrowing priority follows the pecking order theory. Internal resources, like liquidity and equity, are used for investments as long as enough surplus is detected. The surplus of internal resources is the difference between existing internal resources and the expected need for external borrowing on the credit market (see next Section).

## 3.5 Credit market

The credit process embraces the circulation of banking money within the monetary system or the interaction of firms and banks to meet the demand for financing generated by the real

economy. As shown in Fig 3, the credit market has a similar demand and supply structure as the goods market, assuming that banks do not require labor.

**3.5.1 Credit demand.** The investment cycle in Section 3.4 and the credit creation process determine the credit request incidence. Hence, no particular incidence rate is needed. Banks are chosen according to weighted random selection (Fig 5, lines 11,12) when a firm decides to request a loan, based on the interest rate offered (Eq 11). The preconditions (Fig 5, line 4) are based on creditworthiness and regulatory requirement due to Basel III, as described in the next Section.

**3.5.2 Credit supply.** Banks' credit supply is structured into the credit creation process and the corresponding bank strategy to update equity, interest, and risk variables.

*Credit creation.* The credit creation process is based on credit demand generated by firms' investment behavior (see previous Section) and liquidity shortages. Liquidity shortages are characterized by the scenario that firm $i$ cannot hold sufficient liquidity $Liq_i$ for the targeted production. In this case, the firm instantly requests a credit corresponding to its financial need or target liquidity $Liq_i^T$:

$$Liq_i^T = max(W_i^T - Liq_i, 0) \qquad (6)$$

where $W_i^T$ illustrates expected wage costs, dependent on target production (Eq 1) and corresponding labor demand (Eq 3). The credit approval process by banks is subject to two restrictions: credit assessment restriction of firms and regulatory restrictions according to Basel III [42].

Banks' **credit assessment** of firm credit requests evaluates whether the credit demanding firm can fund the loan redemption rate and has sufficient collateral composed of liquid assets (firm liquidity and equity). Following Schasfoort et al. [28], the bank grants a loan if the expected return $r_b^e \geq 0$:

$$r_b^e = \underbrace{\sum_{n=1}^{dur} \frac{\Pi_i^e}{(1+i^L)^n} - L'_{ib}}_{\text{NPV}} - \underbrace{lgd_{ib} \cdot pd_{ib} \cdot L'_{ib}}_{\text{EL}} \qquad (7)$$

The expected return consists of net present value (NPV) and expected loss (EL). The net present value discounts the expected profits $\Pi_i^e$ with the borrowing interest rate $i^L$ over loan duration $dur$ minus the potential loan $L'_{ib}$. The offered interest rate is composed of the key rate (Section 3.6.2) + the fixed standing facility spread of 1% (Section 3.6.1) + margin/mark-up (Eq 11). In line with van der Hoog [24], the loan duration is assumed to be 18 periods. The expected profit is the linear extrapolation of firm turnover over the last three periods plus nominal variation of inventories, deposit interest earnings minus wages, and interest costs [45]. Hence, the NPV reveals the maximum financial absorption capacity for investments (Section 3.4), generating credit purchasing power [11].

The expected loss consists of the loss given default $lgd_{ib}$ and probability of default $pd_{ib}$, concerning risk exposure $L'_{ib}$. The loss given default is the relation between collateral $C$ and risk exposure $L'_{ib}$ and indicates the unsecured credit amount: $\frac{L'_{ib}-C}{L'_{ib}}$. The definition of a bank's expected probability of default $pd_{ib}$ is implemented via a logistic function in line with Caiani et al. [45]:

$$pd_{ib} = \frac{1}{1 + e^{\left(\frac{OCF_i^e - v_b LR'_i}{LR'_i}\right)}} \qquad (8)$$

where $v_b$ indicates the bank's risk attitude and $LR'_i$ depicts the loan redemption rate of the hypothetical loan: $\left(i'_b + \frac{1}{dur}\right)L'$. For simplicity, the interest expenses are distributed equally over the loan period *dur*, which leads to equal loan redemption rates per period. The expected operating cash flow $OCF^e_i$ is the linear turnover extrapolation of the last three periods minus production (wages) and borrowing costs (coupon and interest payments) per period [45].

Here, we follow the interpretation of Caiani et al. [45], that the expected operating cash flow (OCF) indicates a firms' borrowing absorption capacity or "Minskian" litmus paper: a positive OCF indicates excess cash flow to absorb current and upcoming $LR'_i$ (hedge position). A negative $OCF^e_i$ with an absolute value less or equal to the principal repayment still can cover interest costs due and can be classified as a speculative position. Whereby a negative OCF with an absolute value greater than principal payments signals a Ponzi position. The speculative and Ponzi position might lead to credit booms (unproductive credits—*requirement 2*) and severe recessions [37].

The implementation of the expected return has a pro-cyclical effect because expected profits and collateral increase/decrease until borrowing costs and production capacity reach the tipping point.

If the credit assessment is positive, the bank either grants the full loan or only the maximum possible loan amount corresponding to the firm's creditworthiness, thus conducting credit rationing [59]. Firms can utilize several parallel credit lines. If a requested loan is not fully granted by a bank, firms switch their (partial) credit request to another bank. If no bank grants the requested (partial) loan, an overdraft facility can be used which is restricted via debt ratio (Eq 10) and insolvency hurdle (Section 3.3.4). The overdraft facility is only related to loan requests due to liquidity shortages (Eq 6) and functions as a liquidity buffer.

The second credit restriction is related to the **regulatory** framework (*requirement 5*) which determines the regulatory credit creation potential $\tilde{C}_b$:

$$\tilde{C}_b = \frac{E_b}{\bar{\sigma}_b \cdot CAR} \tag{9}$$

where $E_b$ is the equity capital, and *CAR* represents the capital adequacy requirement of 7% according to Basel III [42] (core capital of 4.5% and capital conservation buffer of 2.5%). The parameter $\bar{\sigma}_b$ depicts the risk weighting of all loans for a bank, including the new potential loan $L'_{ib}$. Whereas a solvency rating, according to Basel III regulation [42], defines the weight inbetween 0.2 (high solvency) and 1.5 (low solvency) [29]. The solvency rate $S_i$ (see also Eq 5) is defined as the debt ratio which is derived from the relation between a firm's liabilities (total loans $L$ and overdraft facility $O_i$) and total assets (liquidity $Liq_i$, equity $E_i$ and expected operating cash flow $OCF^e_i$):

$$DR_i = \frac{\sum_{k=1}^{n} L_k + O_i}{Liq_i + E_i + OCF^e_i} \tag{10}$$

According to Basel III [42], the capital requirement is extended by a counter-cyclical buffer $CCycB_b$ and in line with Krug [29]: $CCycB_b = [(G - G^T) - J] \cdot \frac{2.5}{H-J}$. $G$ stands for the current aggregate credit-to-GDP gap and $G^T$ for its long-term trend, which is updated every 12 periods and calculated as the simple moving average over the last 60 periods [30].

A further regulatory requirement is the **leverage requirement**, calculated as the ratio between equity and total assets and required to be above 3%. If a bank does not meet the

capital or leverage requirements, credit creation is discontinued until enough equity is gathered or less risky credit receivables are balanced.

Liquidity constraints according to Basel III (Liquidity Coverage Ratio and Net Stable Funding Ratio) [42] are not implemented as, for simplicity, no active bank liquidity management is assumed.

Banks' credit potential can be increased by accepting low-risk loans or expanding equity, whereas risk weighting is adjusted every period. Combined with firms' pro-cyclical creditworthiness, this mechanism captures the financial accelerator (balance sheet channel—*requirement 6*) because the relative risk weight is evaluated at a lower level in boom phases than in bust phases. Hence, the described price-return cascade of investments (Section 3.4) is fueled by credit supply (purchasing power of credits and potential unsustainable credits—*requirement 2*). Credit potential is expanded pro-cyclically, and regulatory restrictions are circumvented over time due to higher bank profits and corresponding equity (Eq 9). This mechanism produces unproductive credits (*requirement 2*). High expected profits and high collateral disguise the high default risk of firms.

*Bank strategy.* The update rule of banks to adapt to credit demand $CD_b$ and increase the capacity to grant loans and related profits is similar to Fig 4 and Eq 1:

$$
\begin{aligned}
\tilde{C}_b < CD_b \text{ and } m_b \leq \bar{m}_b \Rightarrow E'_b &= E_b + min(Liq_b^A, E_b^N); \\
m'_b &= m_b + \gamma_m \cdot \xi \\
\tilde{C}_b < CD_b \text{ and } m_b > \bar{m}_b \Rightarrow E'_b &= E_b + min(Liq_b^A, E_b^N); \\
v'_b &= v_b \cdot [1 - \gamma_r \cdot \xi] \\
\tilde{C}_b > CD_b \text{ and } m_b > \bar{m}_b \Rightarrow m'_b &= m_b - \gamma_m \cdot \xi \\
\tilde{C}_b > CD_b \text{ and } m_b \leq \bar{m}_b \Rightarrow v'_b &= v_b \cdot [1 + \gamma_r \cdot \xi]
\end{aligned}
\tag{11}
$$

The parameter $\tilde{C}_b$ represents the banks' credit creation potential (Eq 9) and $E_b$ equity capital. The variable $m_b$ depicts the interest margin and $\bar{m}_b$ the average margin of banks at time $t$. The margin is changed stepwise using a calibration parameter $\gamma_m$ and a random variable $\xi$ ($U$ [0, 1]).

Due to regulatory restrictions, banks can only expand credit supply via a higher equity level, which is mainly increased by the retention of profits: $min(Liq_b^A, E_b^N)$. Whereby the variable $Liq_b^A$ defines available liquidity after anticipating liquidity outflow of interbank and Central Bank credit interest charges (Section 3.6.1). $E_b^N$ is the equity need to meet credit demand and credit creation potential in Eq 9. The variable $v_b$ represents the risk attitude of the bank and influences the credit assessment (see Eq 8) and collateral valuation (risk channel—*requirement 6*). A low-risk level equals risk aversion, which devalues firms' collateral and fuels overshooting credit supply in boom phases and vice versa.

The underlying structure of the incidence rate is comparable to line 3 in Fig 4 or the corresponding Eq 2. The banks' production variable is the credit creation potential in Eq 9. The credit creation target is determined by the maximum risk exposure allowed by the regulatory framework [42] with a calibrated buffer parameter. The lower the buffer to the maximum risk exposure, the higher the incidence rate, and vice versa.

**3.5.3 Bank bailout mechanism.** The bailout mechanism defines how the government recapitalizes insolvent banks. Insolvent banks exhibit negative equity, caused by non-performing loans from bankrupt firms, as described in Section 3.3.4. The government has no debt restrictions in the model and can intervene anytime and with any amount. The bailout

mechanism is related to Popoyan [30] and Krug [29] (see Fig 3): It is assumed to keep the number of banks constant. The bailout sum depends on losses from non-performing loans and minimum capital requirements due to Basel III regulation (Section 3.5.2). Banks get at least the minimum amount of equity to fulfill capital requirements.

### 3.6 Monetary system

While the banking money circuit is similar within alternative monetary regimes, the institutional framework of the interbank circuit is different. Mak(h)ro_0 incorporates the current monetary system, also known as the fractional reserve system.

**3.6.1 Interbank market.** The interbank market defines the interaction between banks and the Central Bank to guarantee payment settlements of economic transactions. In co-evolution with the credit market the interbank market distinguishes between banking and central bank money. Banking money circulates when endogenously issued by credit supply to firms (Section 3.5 and *requirement 4*). Hence, deposits are created when banking money is issued directly to firms (when borrowing) and indirectly to households (via wages, when employed with external financed investments).

All economic transactions (e.g., household consumption, wage payments, investment purchases, etc.) are settled via deposit accounts of the respective bank (see S2 Table). According to the monetary policy operation framework [43, 63, 64], transactions across banks require central bank money (reserves). The settlement of all transactions across all banks is assumed to be twenty times within a period and is instantly balanced with reserves. However, reserves are constantly unevenly distributed due to heterogeneous agent behavior (more/less transaction volume for a specific bank). Therefore, banks with a shortage of reserves can borrow from banks with excess reserves on the interbank market at the end of each settlement period. This system is also called a fractional reserve system because only a fraction of reserves is required to let money circulate (*requirement 3*). The oversupply (excess reserves $RE_b$) or undersupply (reserves demand $RD_b$) of reserves is indicated by the interest rate for interbank loans $im_b$:

$$
\begin{aligned}
RE_b = RD_b \text{ and } im_b > \overline{im_b} &\Rightarrow im'_b = im_b - \gamma_i \cdot \xi \\
RE_b = RD_b \text{ and } im_b < \overline{im_b} &\Rightarrow im'_b = im_b + \gamma_i \cdot \xi \\
RE_b < RD_b &\Rightarrow im'_b = im_b + \gamma_i \cdot \xi \\
RE_b > RD_b &\Rightarrow im'_b = im_b - \gamma_i \cdot \xi
\end{aligned}
\tag{12}
$$

The market mechanism is similar to the update rule of Eqs 1 and 11, where $\gamma_i$ is a calibration parameter, and $\xi$ a random variable ($U[0, 1]$). The incidence rate is assumed to be 20, which means that the strategy is conducted twenty times on average within a period. The match between excess and deficit reserve banks is implemented based on the matching protocol in lines 11 and 12 in Fig 5. The interbank loan request is similar to the firm credit market in Section 3.5, whereas the creditworthiness of other banks is evaluated based on their solvency. If the bank that requests an interbank loan does not have sufficient equity to fulfill the Basel III requirements, no interbank loan will be granted, and the Central Bank has to intervene as Lender of Last Resort.

*Open market operations.* Further unbalancing effects of payment transactions and the corresponding need for reserves are caused by the credit creation process from banks to firms (Section 3.5). Higher credit amount/deposits in the economy lead to new/more transactions and vice versa. However, when banks cannot gather enough reserves from other banks, the Central Bank provides a marginal lending facility (short-term credit line). Short-term is defined with

an incidence rate of 20, as described above. On the other hand, when too many reserves are circulating, they are posted to the deposit facility by the Central Bank, which is an interest-bearing account. The marginal lending and deposit facility (standing facilities) are set at 100 basis points above/below the key rate (main refinancing operations rate) as it reflects the European Central Bank strategy prior to the financial crisis of 2007/2008. Hence, the interbank rate will always be kept between the standing facilities.

To keep the interbank rate close to the key interest rate and the corresponding volatility low, the Central Bank intervenes via open market operations. It is assumed that the Central Bank provides/withdraws reserves when the average interbank rate diverges from the key rate. The higher the deviation from the key rate, the more frequent the intervention.

*Maintenance period and reserve target.* Banks have to hold a minimum reserve requirement to stabilize the interbank rate, which means they need a fraction of reserves in relation to their deposits. The fraction is set at 1% [64] and needs to be held on average over the reserve maintenance period of every two periods. Banks can also use those fixed reserves, during that time, which dampens ad hoc reserve demand within the interbank market and stabilizes the interbank rate. It is assumed that banks are willing to close the minimum reserve shortage gap as soon as possible, reflecting the fact that banks face uncertainty regarding interest changes and the ability to gather enough reserves to fulfill the reserve requirement. The model does not include autonomous factors that influence banks' liquidity like cash requirements and government deposits.

The reserve target follows the maintenance period and an expected reserves outflow based on the latest average firm investments, household consumption, and firm wage transactions.

Open market operations or Central Bank credits are empirically based on banks' securities/collateral, such as covered bonds, asset-backed securities, etc. Here we assume an unsecured interbank market. Banks have no lending restrictions concerning reserves [41], because the Central Bank will keep the interbank rate close to the key rate and always provides sufficient liquidity (see fine tuning operations [43]). It is assumed that endogenous interbank crashes are very scarce since the Central Bank always acts as a Lender of Last Resort and keeps enough required liquidity to dampen interbank rate fluctuations. Hence, an interbank shutdown can only be triggered by an exogenous shock which generates high interbank market imbalances and leads to credit supply stops and a disruption in the payment mechanism. When conducting experiments, an interbank specification can be activated in the model to measure liquidity risk via interbank risk premia and exploitation of Central Bank standing facilities, which can impede credit supply costs. Bank runs are not assumed.

*Bank deposit strategy.* To implement changing bank relationships and the desire to attract reserves, households change their bank account on average every 12 periods and firms every 100 periods. The deposit interest rate determines the incentive for households and firms to invest their deposits at a particular bank. The deposit margin/mark-up is adapted according to demand via an equivalent mechanism in Eq 12. The selection process for depositors is based on the known matching protocol (lines 11, 12 in Fig 5).

*Interest spreads.* The spread of the interest corridor of the standing facilities is assumed not to change with interest level changes. The assumption holds true for the interest spreads between several interest types. The offered loan interest rate to firms by banks is composed of the key rate plus the fixed standing facility spread of 1% and the individual bank loan mark-up (Eq 11). Banks' deposit rate offered to depositors comprises the key rate minus the fixed standing facility spread of 1% and a calibration parameter of 3% plus the individual bank deposit mark-up.

**3.6.2 Monetary policy mechanism.** As already described in the previous Section, monetary policy is based on inflation targeting, according to Bindseil [64]. The Central Bank

measures the average inflation over 12 periods and compares the value with the inflation target of 2%. The inflation target influences the general interest level within the model and is set based on a corresponding calibration process concerning stylized facts (see next Section and Section 5). Furthermore, monetary policy considers the output gap, implemented via the original Taylor rule [65]:

$$i_c = i^* \ + \ a_\pi(\pi - \pi^*) \ + \ a_\pi(y - y^*) \tag{13}$$

where $i_c$ is the key rate, $\pi$ the inflation rate measured over the last 12 periods, $\pi^*$ the inflation target and $y-y^*$ the real output gap in relative terms. The potential real output $y^*$ is assumed to be 95% of full production (i.e., all households are employed) which is in line with the Non-Accelerating Inflation Rate of Unemployment (NAIRU) in advanced economies (see Section 3.3.1). Within the model, 5% serves as a buffer for inflationary pressure which means that the output gap can become temporarily positive. One of the main components in the model is the natural interest parameter $i^*$ in Eq 13, which is calibrated at 10%.

## 3.7 Fiscal mechanism

The government raises taxes to finance unemployment benefits and bank bailout measures. The tax is applied to wages and firm/bank profits. Since firm profits are part of the creditworthiness, taxes impact the model behavior. They are calibrated in combination with the Taylor rule in the previous Section that defines the general interest level. If households become unemployed, the government provides a tax-exempt unemployment benefit calibrated at half of the fixed wage. Bank bailouts are described in Section 3.5.3. All bankruptcy costs are covered by the government, which has no liquidity limit.

## 4 Calibration

The model input refers to parameter values as well as the initial values of variables, for a complete list see the supplement [66]. One possibility to determine suitable values is their calibration [13].

### 4.1 Parameter calibration

As shown in Fig 3, the model is composed of various submodels. The submodels "Goods market", "Investment market", "Financial" sector and the final model composition have undergone a sensitivity analysis based on theoretical and empirical plausibility with a one-factor-at-a-time (OFAT) approach [67]. Furthermore, during the model building process, a Plackett-Burman experimental design has been applied to the submodels "Goods market" and "Investment market" to identify parameter limits in line with stylized behavior. The monetary policy specification is only calibrated with the final composite model. Since no productivity growth is assumed, the calibration of the final model is based on an aggregate credit amount that is allowed to fluctuate but should not show a growth trend.

### 4.2 Initialization

As stated in Fagiolo and Roventini [13], complex stochastic models are characterized by high levels of dimensionality, leading to the circumstance that the statistical equilibria are unknown to the modeler. To keep that problem under control, we adopt the approach of Dosi et al. [54] to choose "economically meaningful variables". Initial values are anticipated from the average production capacity for each firm. The average production capacity per firm is the total number of households divided by the total number of firms. The initial demand sources,

households (wages), and firms (investment amount), in combination with bank credit creation potential (bank equity), are initialized with adequate values to let firms emerge to their full production capacity in an aggregate perspective. All other related values (inventories, liquidity, equity of firms) are calibrated/iterated according to their respective average values in the model output. Initial economic values affect the length of the warm-up but not the model behavior. Hence, "adequate" values refer to a set of values that minimize the impact of the warm-up phase, which can be appraised between 100 and 200 periods. Furthermore, all agents are initialized homogeneously. The heterogeneous agent structure emerges during the simulation via the stochastic feature of ML3.

The size of each agent (sub-)population is selected in a manner that its scale to other sub-populations appears plausible following model assumptions (Section 3.2.1) and other MABM [24, 29, 30, 45]. For each bank, we assume four times as many firms [68] and 40 times as many households (e.g., with 25 banks, 100 firms, and 1000 households).

A change of agent size relations comes with an adaption of the calibration to replicate stylized facts (see next Section) but does not change the fundamental model behavior. However, a constant relation between agent sizes does not affect the model behavior between population size variations of 10 to 200 firms. Too small population sizes might affect simulation results due to stochasticity. With the current simulator, more than 200 firms cannot be tested due to technical restrictions. The validation is conducted with a population size of 12 banks, 48 firms and 480 households, and one Central Bank and government. The simulation is run for 600 periods of simulated time.

## 5 Validation

The purpose of the model is to analyze the effects of institutional system changes or policy experiments in terms of macro-financial stability. A common method to test whether the model output can be used to identify policy experiment effects is "to reproduce jointly a wide range of macroeconomic and microeconomic stylized facts" [69]. A "history-friendly approach" [22] that describes a strict adaptation to historical data would not be recommendable for the purpose of the present model due to the methodological pitfall of counter-factual histories.

The validation is based on a selected set of macro and microeconomic stylized facts (SF) suggested by Dosi et al. [69] (see Table 1), which is considered the most comprehensive set of stylized facts in the macro-finance realm [29]. This set mainly replicates stylized facts from OECD economies, specifically the U.S., making the validation comparable to other MABM. Long-term growth-related stylized facts are not under examination (see the discussion of model assumptions and limitations in Section 3.2.1). Compared to Dosi et al. [69], we combined SFs (e.g., SF1 and SF2 in [69]), excluded SFs related to long-term growth or productivity, and expanded SFs w.r.t. the monetary system (e.g., Phillips curve and interbank market).

To validate the model output following stylized facts, the micro-founded model outcome has to be aggregated. An important economic output indicator is the Gross Domestic Product (GDP) which measures the aggregate value of all finished goods and services in a certain period of time within an economy. Here, (real) GDP is defined as the aggregated quantity of final goods produced by firms for household consumption $C$ and firm investment $I$ plus change in inventories $\Delta N_i$ in production units $q$ (homogeneous good) within a period [69]:

$$GDP^{real} = \sum_{i=1}^{n} q_i^C + \sum_{i=1}^{n} q_i^I \equiv C + I + \Delta N_i \tag{14}$$

**Table 1. Stylized facts replicated by the model.**

| Code | Stylized fact | Empirical studies (examples) |
| --- | --- | --- |
| SF1 | Relative volatility of GDP/invest./consum. | Stock and Watson [70]; Napoletano et al. [71] |
| SF2 | Recession duration exponentially distributed | Ausloos et al. [72]; Wright [73] |
| SF3 | Cross-correlations of macro variables | Stock and Watson [70]; Napoletano et al. [71] |
| SF4 | Pro-cyclical aggregate R&D investment | Wälde and Woitek [74] |
| SF5 | Cross-correlations of credit-related variables | Lown and Morgan [75] |
| SF6 | Lagged correlation between firm indebtedness & credit defaults | Foos et al. [76]; Mendoza and Terrones [77] |
| SF7 | Banking crises duration is right skewed | Reinhart and Rogoff [78] |
| SF8 | Fat-tailed distribution of fiscal costs of banking crises-to-GDP ratio | Laeven and Valencia [2, 79] |
| SF9 | Lumpy investment rates at firm-level | Doms and Dunne [80] |
| SF10 | Firm bankruptcies are counter-cyclical | Jaimovich and Floetotto [81] |
| SF11[a] | Phillips curve | Phillips [82] |
| SF12 | Okun's law | Okun [83] |
| SF13 | Interbank market | BIS [84] |

[a]Described as a general characteristic of an economy, i.e., without an explicit notion of empirical studies, and found in Riccetti et al. [85], Krug [29] and Delli Gatti et al. [18].

Nominal GDP is measured similarly to Eq 14 with firm prices. It is worth mentioning that $\Delta N_i$ is not part of $I$ (see Section 3.4). To extract the "business cycle component" of a given time series variable within a particular band of frequencies, we use the Baxter-King filter with standard configurations for economic time series. Stock and Watson [70] state that business cycles last from 6 to 32 quarters which specifies the lower and upper bound of the cycle periodicity. The filter's fixed lead/lag length or order is set at 12. These configurations are used for quarterly data in reality and are indicated with the following notation: bpf(6,32,12). Our model is calibrated such that one period corresponds to a quarter in reality, thus, 400 periods correspond to 100 years. All validation outputs are based on the same parameter and variable setting. The validation is based on 20 replications, which can be interpreted as representative since the average real GDP standard deviation across all 20 replications is very small ($\rho$ = 3.471e-04). However, for future work, when interpreting monetary regime changes, the number of replications has to be determined using a transient analysis and an appropriate confidence level. All analyses are conducted in R and can be found in the supplement [66].

## 5.1 Relative volatility of GDP, investment and consumption—SF1

The first stylized fact depicts the behavior at business cycle frequencies related to GDP and firm investment time series. Fig 6 shows that the model output can replicate empirical findings in the sense that investment fluctuations are more volatile than GDP time series [70, 71]. To keep the graph uncluttered, consumption is not included and can be found in S1 Fig, which shows a less volatile behavior. Using 20 simulation replications, the relative standard deviation compared to real GDP of real consumption is 0.44, 4.58 for real investment and 13.51 for unemployment. Empirical observations [86] show a relative standard deviation compared to real GDP of 0.80, 4.61 for investment, and 8.22 for unemployment. As described in Section 3.4 consumption is crowded out by investment activities which leads to a lower consumption

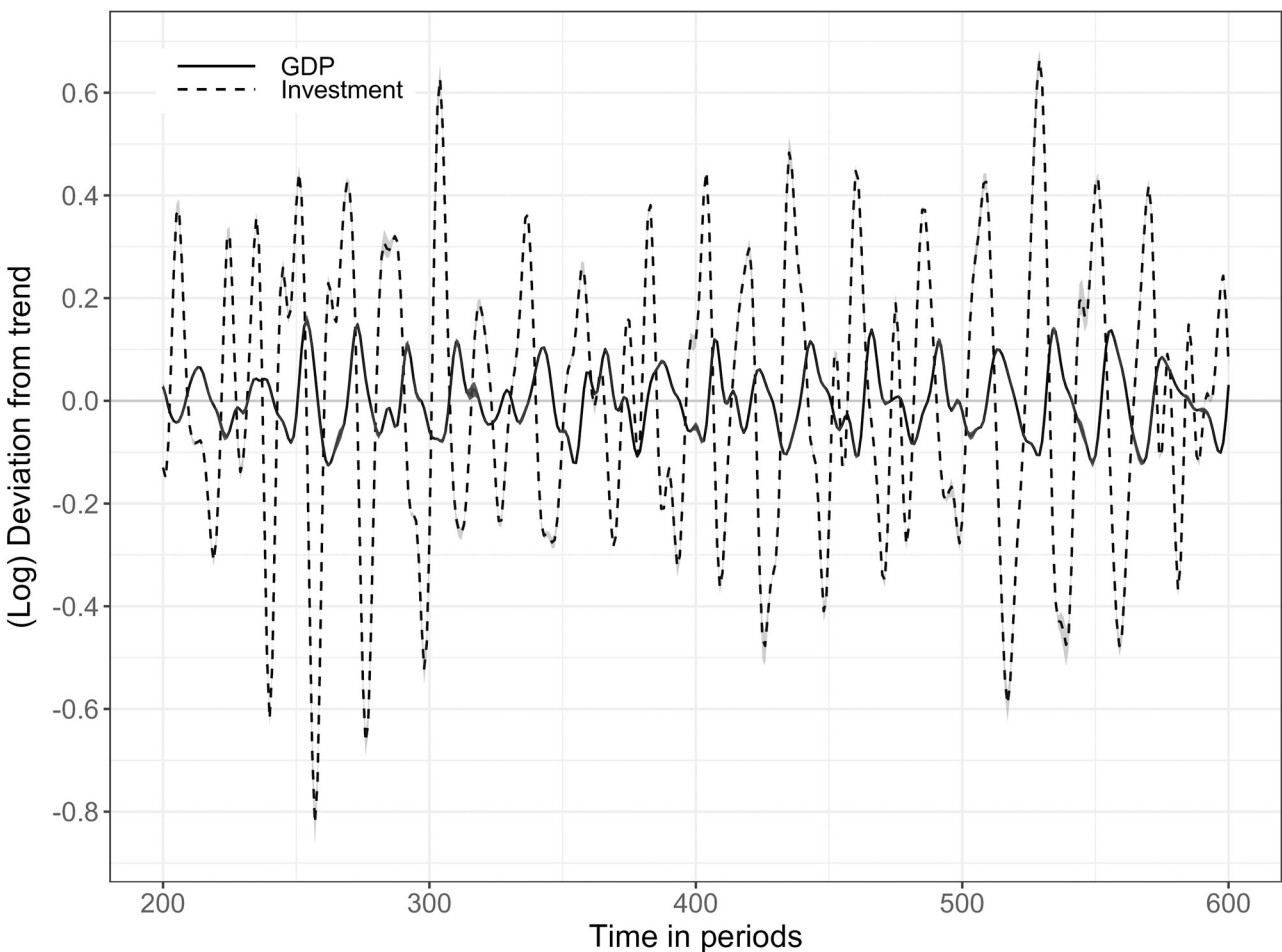

**Fig 6. SF1: Deviations of real GDP and real investment time series from trend: Bandpass-filtered (6,32,12). Note:** The Figure depicts deviations of average log real GDP (solid line) and average log real firm investment (dashed line) from trend. The simulation output is a representative run and shows an extract to keep the line profile more identifiable without the warm-up phase of 200 simulation periods. The grey shaded area represents the 90% confidence interval.

volatility. A calibration with a higher government tax rate (e.g., tax = 0.13) lowers the relative unemployment standard deviation and consumption, but keeps investment at around 4.60. In addition, the model output can replicate persistent fluctuations [70, 87, 88] but no long-term growth due to model assumptions (see Section 3.2.1).

### 5.2 Recession duration exponentially distributed—SF2

The recession duration is defined as the length of periods in which real GDP growth percentage change is less than zero [73]. The model output is presented in Fig 7 and shows a clear exponential distribution, which is in line with empirical data [72, 73].

### 5.3 Cross-correlations of macro variables—SF3

The third stylized fact that can be replicated by the model is related to several macro variables which are presented in Table 2. Investment, change in inventories and inflation are pro-cyclical and unemployment counter-cyclical which is in line with Wälde and Woitek [74].

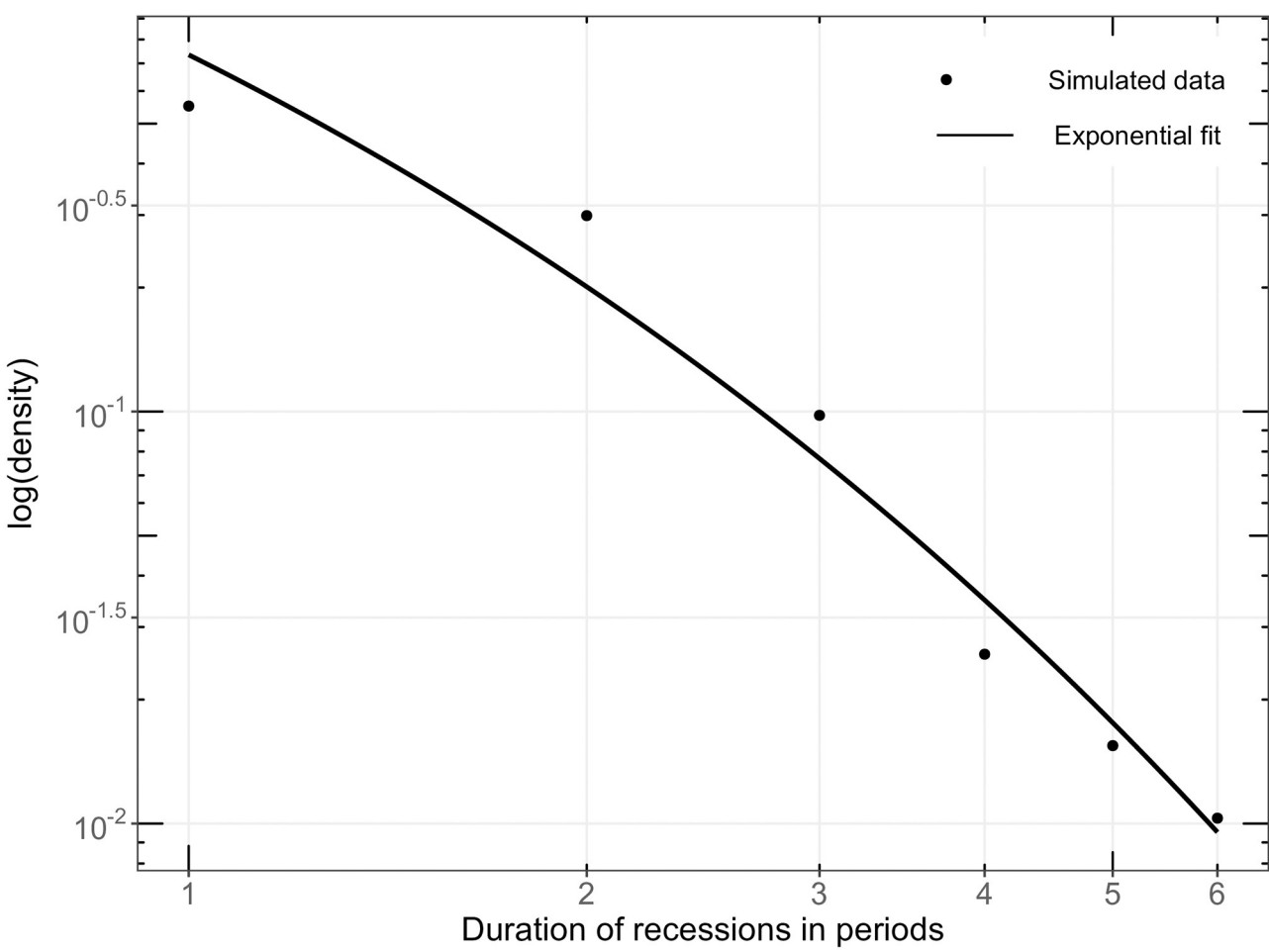

**Fig 7. SF2: Exponential fit of recession durations. Note:** Graphical analysis of the frequency of recession duration plotted as a cumulative probability distribution in log-log scale according to Wright [73] and Dosi et al. [69]. The points present the simulated outcome and the solid line depicts the exponential fit to the data. The recession duration is defined as the length of periods in which real GDP growth percentage change is less than zero. The simulation output is a representative run without the warm-up phase of 200 simulation periods.

Investment is leading real GDP, because investment demand is satisfied by firms' inventory capacities and not by current production capacity, as demonstrated with an alternative GDP measure in Table 2 (GDP −Δ inventories). This circumstance can be adapted by introducing a separate agent type for investments, but this in turn makes the model more complex.

Changes in inventories lag real GDP because of the gradual build-up of production capacity (Section 3.3.1). Demand impulses cause an adaption of the production target which in turn activates the hire/fire mechanism (Section 3.3.2) and finally the production capacity. Inflation is lagging GDP by up to three lags, because inflation is measured over three periods. In contrast to Wälde and Woitek [74] prices are pro-cyclical. However, this is in line with a monetary or demand-driven economy [89] and thus with the purpose of the model to compare monetary systems. Counter-cyclical prices are caused by supply shocks. Due to fixed wages and productivity, a supply shock is not incorporated, but can be implemented via external shocks.

**Table 2. Cross-correlation of macro variables—SF3.**

| Series (Bpf) | Output (Bpf) | | | | | | | | |
|---|---|---|---|---|---|---|---|---|---|
| | t−4 | t−3 | t−2 | t−1 | t | t+1 | t+2 | t+3 | t+4 |
| **GDP** | 0.189 | 0.465 | 0.731 | 0.927 | 1 | 0.927 | 0.731 | 0.465 | 0.189 |
| Investment | 0.516 | 0.318 | 0.073 | -0.191 | -0.438 | -0.635 | -0.755 | -0.788 | -0.738 |
| Change in inventories | -0.271 | 0.002 | 0.302 | 0.578 | 0.779 | 0.872 | 0.854 | 0.746 | 0.579 |
| Inflation | 0.004 | 0.19 | 0.365 | 0.515 | 0.626 | 0.687 | 0.689 | 0.626 | 0.502 |
| Prices | 0.418 | 0.561 | 0.663 | 0.707 | 0.679 | 0.575 | 0.402 | 0.183 | -0.052 |
| Unemployment | -0.353 | -0.555 | -0.724 | -0.822 | -0.819 | -0.706 | -0.5 | -0.242 | 0.023 |
| **GDP** −Δ inventories[a] | 0.06 | 0.349 | 0.66 | 0.906 | 1 | 0.906 | 0.66 | 0.349 | 0.06 |
| Investment | 0.084 | 0.364 | 0.659 | 0.892 | 0.985 | 0.903 | 0.673 | 0.37 | 0.077 |

[a]Cross-correlation between investment and GDP measurement without change of inventories.

Correlation structure of simulated macro variables to real GDP.

The correlation coefficients are related to the dependent variable GDP. The simulation output comprises 20 representative runs to increase the sample with a total number of 12,000 period observations without the warm-up phase of 200 simulation periods.

Bpf: bandpass filtered (6,32,12) series.

## 5.4 Pro-cyclical aggregate R&D investment—SF4

Aggregate investments of firms per period are pro-cyclical which is shown graphically in Fig 8. As described in the previous Subsection, investment is characterized by a leading feature relative to real GDP.

## 5.5 Cross-correlation of credit variables—SF5

Lown and Morgan [75] indicate that bank profits and total debt outstanding in the firm sector are pro-cyclical. The related model output presented in Table 3 can confirm these empirical findings. Bank profits are lagging real GDP by one lag, because the loan redemption starts one period after the loan was granted.

Firm debt includes bank credits for investments and liquidity injections as described in Section 3.5.1. The concurrent behavior to real GDP shows that demand and supply in the goods market is dependent on the financial absorption capacity.

Credit defaults of firms can be interpreted as counter-cyclical, which is in line with empirical findings [75]. With increasing GDP, credit defaults gradually decline and vice versa. Moreover, credit defaults concur with investment activities (Table 2) that define the business and credit cycle (Section 3.4). With negative correlation of investment, loan losses start to increase. An investment demand decrease causes lower (expected) profits (Eq 7) and cash flow, leading to lower debt-bearing capacity and credit defaults. This effect becomes more severe with increasing indebtedness (see Minskian litmus paper in Section 3.5.2).

## 5.6 Cross-correlation between firm debt and loan losses—SF6

The time-series correlation of aggregated loan losses in relation to aggregated firm debt is positive and lagged, which is demonstrated in Fig 9 and in line with Mendoza and Terrones [77]. Firm debt is mainly driven by bank credit borrowing. High firm debt accumulation of firms turns into a leverage cycle and will burden banks' credit potential [76].

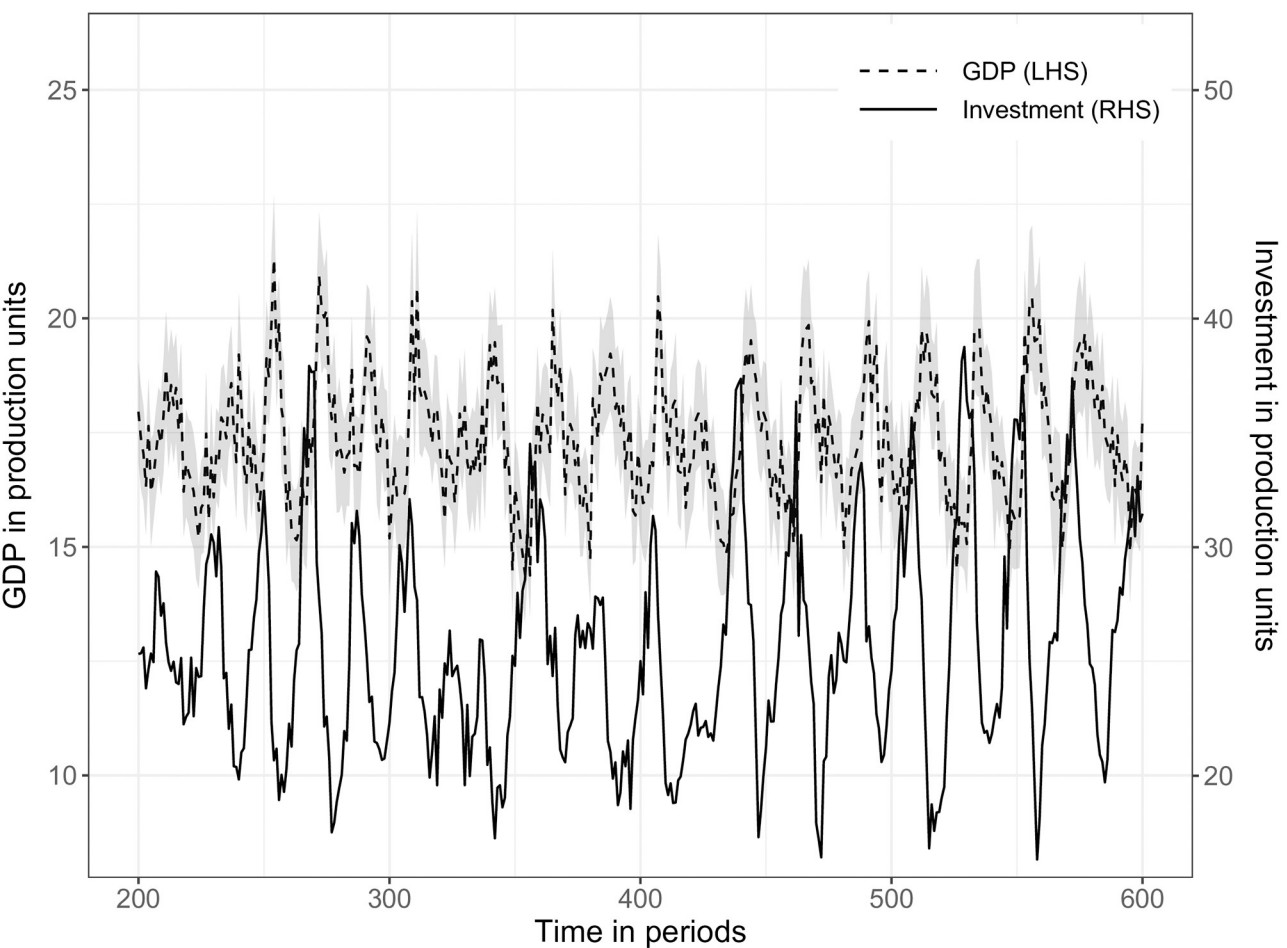

**Fig 8. SF4: Pro-cyclical aggregate firm investment. Note:** The Figure depicts average real GDP as a dashed line and aggregated total investment as a solid line. The simulation output is a representative run and shows an extract to keep the line profile more identifiable without the warm-up phase of 200 simulation periods. The grey shaded area represents the 90% confidence interval.

**Table 3. Cross-correlation of credit variables—SF5.**

| Series (Bpf) | Output (Bpf) | | | | | | | | |
|---|---|---|---|---|---|---|---|---|---|
| | t−4 | t−3 | t−2 | t−1 | t | t+1 | t+2 | t+3 | t+4 |
| **GDP real** | 0.189 | 0.465 | 0.731 | 0.927 | 1 | 0.927 | 0.731 | 0.465 | 0.189 |
| Bank prof. | -0.04 | 0.229 | 0.493 | 0.716 | 0.864 | 0.915 | 0.865 | 0.731 | 0.539 |
| Firm debt | 0.085 | 0.225 | 0.347 | 0.435 | 0.473 | 0.46 | 0.402 | 0.315 | 0.213 |
| Credit def. | -0.002 | -0.012 | -0.024 | -0.023 | 0.001 | 0.039 | 0.073 | 0.084 | 0.064 |

Correlation structure of simulated credit variables to real GDP.

The correlation coefficients are related to the dependent variable GDP. The simulation output comprises 20 representative runs to increase the sample with a total number of 12,000 period observations without the warm-up phase of 200 simulation periods.

Bpf: bandpass filtered (6,32,12) series.

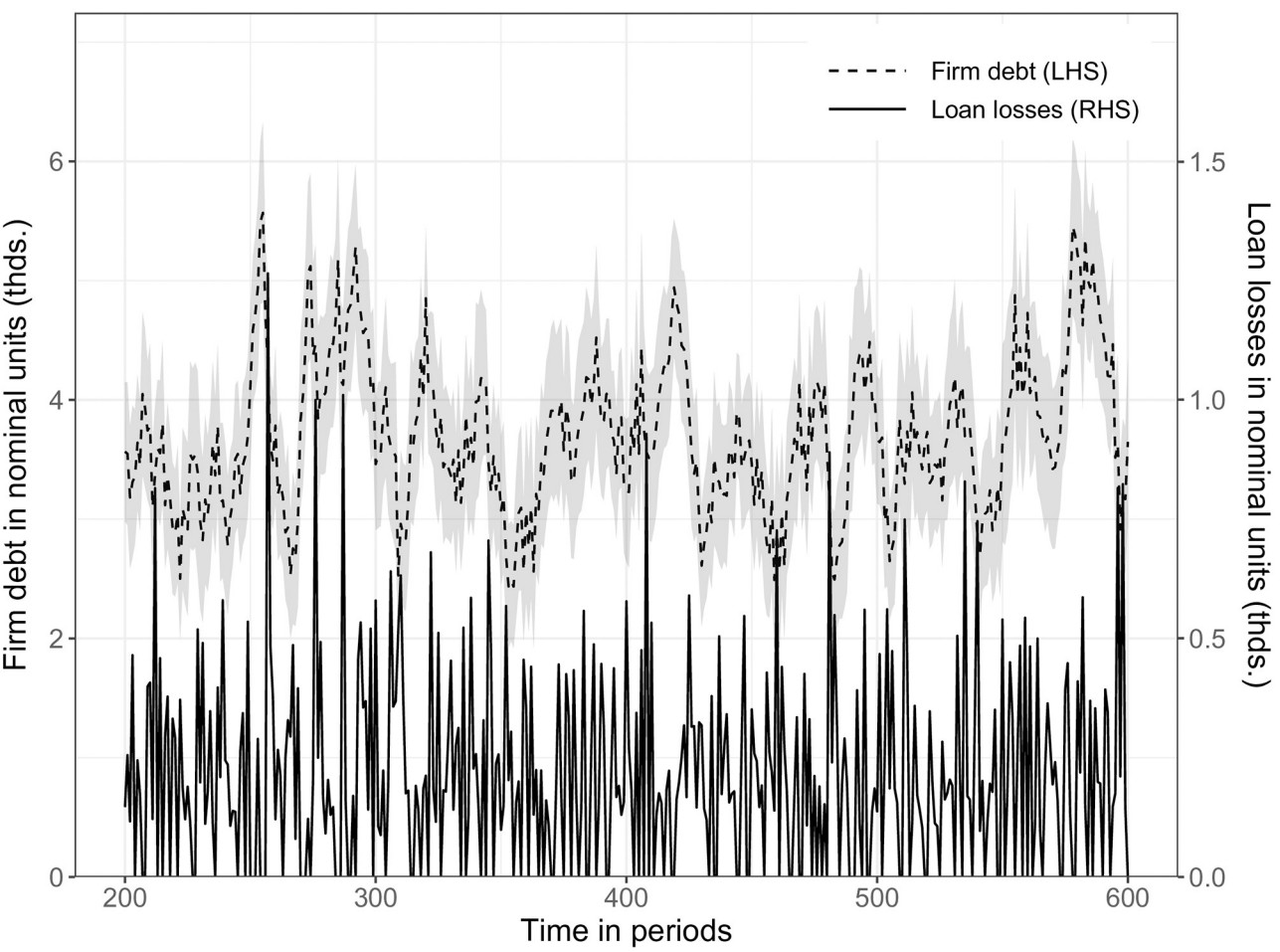

**Fig 9. SF6: Firm debt and loan losses time series. Note:** The Figure depicts average firm debt as a dashed line and average loan losses as a solid line. The simulation output is a representative run and shows an extract to keep the line profile more identifiable without the warm-up phase of 200 simulation periods. The grey shaded area represents the 90% confidence interval.

### 5.7 Banking crisis duration is right-skewed—SF7

Banking crisis duration is defined as the length of periods with bank failures in a row. According to Reinhart and Rogoff [78] the distribution of bank failures duration is right skewed. The model can replicate empirical observations, shown in the second column of Table 4.

### 5.8 Fiscal cost of banking crisis to GDP distribution is fat-tailed—SF8

In relation to the previous Subsection, banks are bailed out by the government. The corresponding ratio of fiscal costs-to-GDP of banking crises duration has a distribution with the feature of excess kurtosis and heavy tails as presented in the third column of Table 4. This observation is in line with empirical findings according to Laeven and Valencia [2, 79]. A Jarque-Bera test for normality yield a p-value of 0.00 and a test statistic of 1922.3, indicating that a normal distribution has to be rejected. The same holds true with a Shapiro-Wilk test.

**Table 4. Distribution related stylized facts.**

|  | SF7: Banking crisis duration | SF8: Fiscal costs/GDP ratio |
|---|---|---|
| Skewness | 1.227 | 1.345 |
| Kurtosis | 1.881 | 3.676 |
| Mean | 7.032 | 0.146 |
| Standard dev. | 6.813 | 0.082 |
| Min | 1 | 6.124e-04 |
| Max | 49 | 0.680 |
| Median | 4 | 0.135 |
| Mode | 1 | 0.017 |

The simulation output comprises 20 representative runs to increase the sample with a total number of 12,000 period observations without the warm-up phase of 200 simulation periods.

### 5.9 Lumpy investment rates at firm-level—SF9

Lumpy investments are infrequent and large (lumpy) establishment-level capital adjustments [90]. Fig 10 depicts the investment behavior per period of three randomly chosen firms. It can be observed that investments do not follow a smooth behavior, but rather indicate periods with successive spikes and low or even no investment activity (Section 3.4.1).

### 5.10 Firm bankruptcies are counter-cyclical—SF10

Empirical findings according to Jaimovich and Floetotto [81] state that firm bankruptcies are counter-cyclical. As shown in Fig 11 the model is able to replicate the counter-cyclical behavior. Firm bankruptcies are leading by two lags in relation to real GDP (with $\rho = -0.230$; band-pass filtered (6,32,12)). The leading feature indicates the beginning of the tipping point between boom and bust states. Firm bankruptcies are dependent on solvency or leverage (Eq 10). In boom phases investments and thus prices increase until the maximum financial absorption capacity. This process is influenced by higher interest rates level (monetary policy—Section 3.6.2), full production capacity and credit rationing (Section 3.5.2).

### 5.11 Phillips curve—SF11

The model is able to replicate a standard Phillips curve which is shown in Fig 12. The time measurement is based on inflation, which is measured every three periods in the model.

### 5.12 Okun's law—SF12

Okun [83] finds a negative statistical relationship between real GDP growth and unemployment rate which can be replicated by Mak(h)ro_0 ABM as shown in Fig 13. Okun's law has been empirically re-investigated several times afterwards and can be derived in various ways (e.g., Rahman and Mustafa [91] and Obst [92]). For simplicity we show the "difference version" which relates to changes in the unemployment rate to changes in real GDP per period. The "gap version" is tested as well and unveils very similar results.

### 5.13 Interbank market—SF13

Concerning the interbank market behavior, Mak(h)ro_0 can replicate three stylized facts according to Bech and Monnet [84]. Here, we conduct an experiment and let the Central Bank inject 10 reserve units per period for every bank between period 50 and 100. The periods

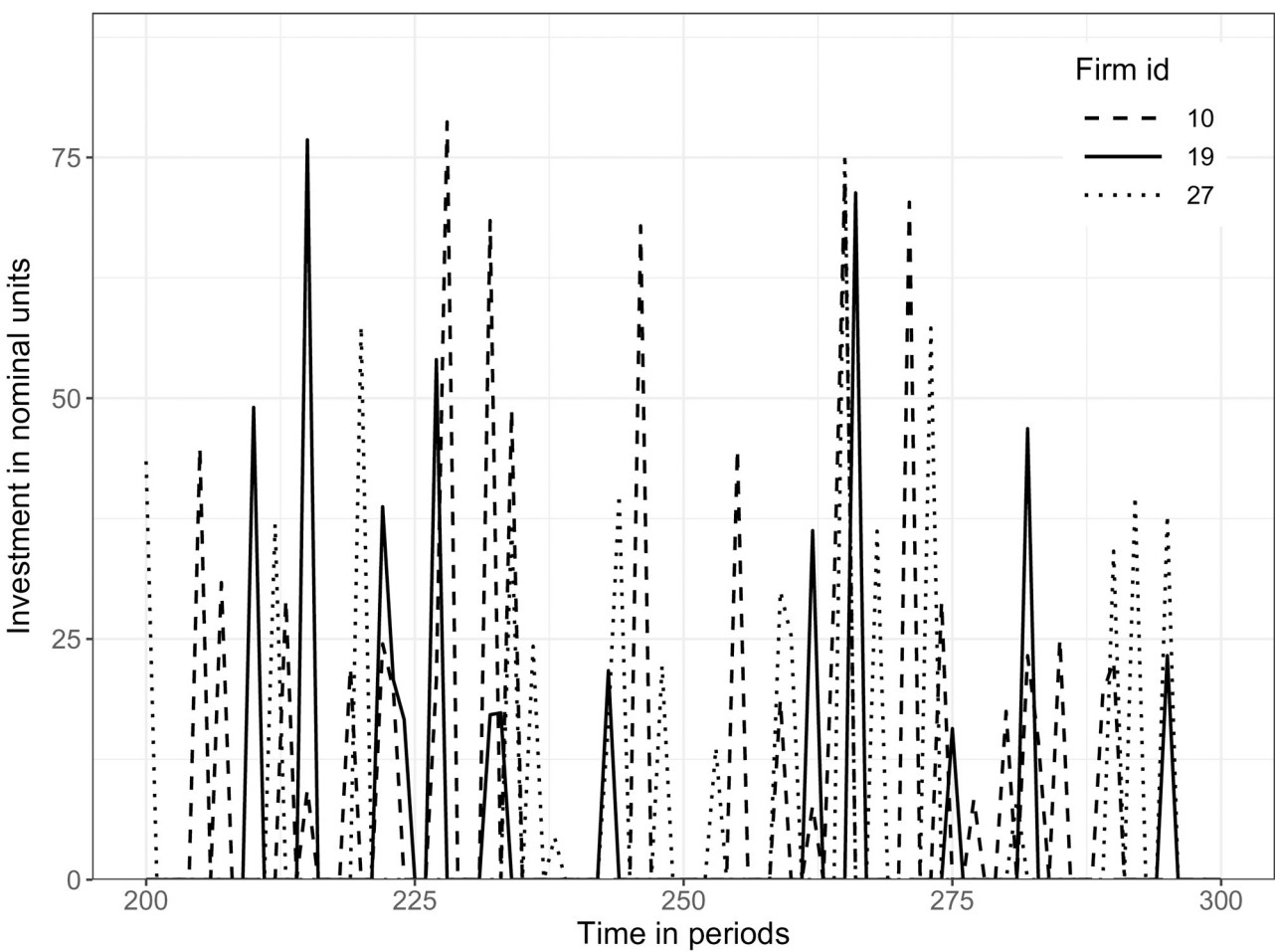

**Fig 10. SF9: Individual firm investment time series. Note:** The Figure depicts investment frequencies for three randomly chosen firms. The dashed line shows the firm with identification number (id) 10, the solid line the firm with id 19 and the dotted line the firm with id 27. The simulation output is a representative run and shows an extract to keep the line profile more identifiable without the warm-up phase of 200 simulation periods.

before and after the experiment phase depict normal open market operations by the Central Bank (Section 3.6.1). The key rate is assumed to be 1%, with an interest corridor of ±1%.

Fig 14 shows, that *increasing the amount of excess reserves drives the overnight interbank rate towards the floor of an interest rate corridor* [84] between period 50 and 100. Second, *increasing the amount of excess reserves reduces overnight interbank volume* [84]: average interbank volume between period 50 and 100 (29.299 reserve units) are significantly lower (p-value: 0.003) than the periods before (53.805 reserve units). Third, *increasing the amount of excess reserves lowers overnight rate volatility* [84]: average standard deviation between period 50 and 100 ($\sigma$ = 8.374e-04) is significantly lower (p-value: 0.000) than the periods before ($\sigma$ = 2.038e-03). Here, we do not include the impact of excess reserves on the recourse to the Central Bank lending facility, since we assume that banks with excess reserves are always willing to lend.

## 6 Discussion and extended literature review

As shown above, the model at hand fulfills all defined requirements in Section 3.1. To show the additional benefit of **this model** in comparison **to other** related MABMs, Table 5 gives a

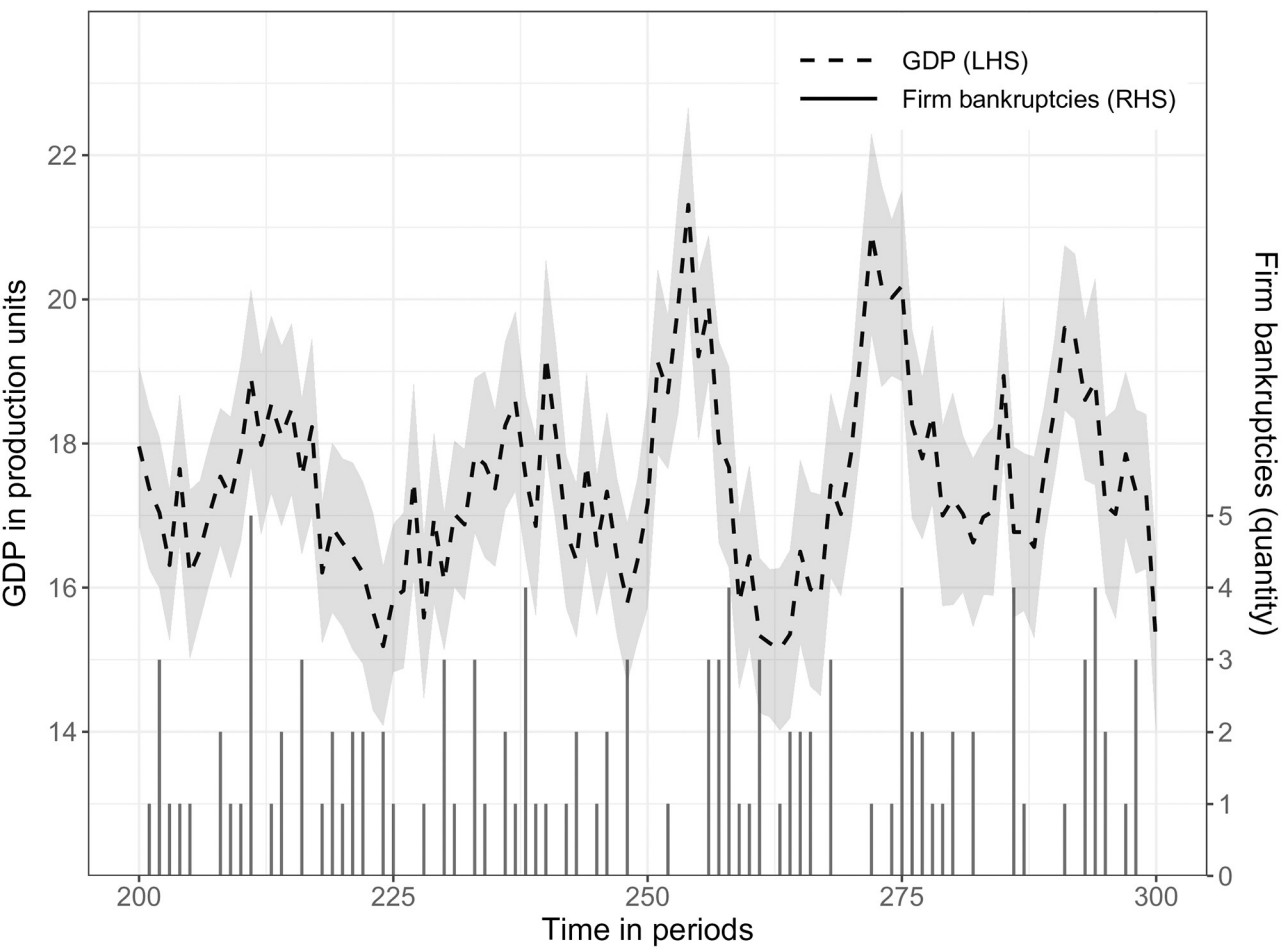

**Fig 11. SF10: Real GDP and firm bankruptcies time series. Note:** The Figure depicts average real GDP as a dashed line and firm bankruptcies as a solid line in bars. The simulation output is a representative run and shows an extract to keep the line profile more identifiable without the warm-up phase of 200 simulation periods. The grey shaded area represents the 90% confidence interval.

concise overview according to the model requirements. Not following all specified model requirements can lead to biased results. However, it should be noted that the purpose of the related MABMs in Table 5 is analyzing various economic policy experiments rather than the monetary system or (alternative) options of monetary regime shifts. Whereby parts of the current monetary system are modeled as a source for bank liquidity constraints [30] and monetary policy transmission [28, 29], e.g., to grasp macroprudential policy changes in interdependence with price stability [29, 30] or the influence of monetary policy transmission on price stability [28]. A first modification of the monetary system has been examined by van der Hoog [24] (more details below and in the introduction). For a more general and comprehensive overview about MABMs the reader is referred to Dawid and Delli Gatti [48] who provide an extensive comparison. The following paragraphs refer to the model requirements defined in Section 3.1:

**The business cycle (** *requirement 1*) described in Section 3.3 and its behavior shown in Section 5.1 to 5.4 and 5.10 to 5.12 is not particular different to related MABMs in Table 5. However, the source of credit demand is modeled differently which has an influence on the credit cycle as required and described in the next paragraph.

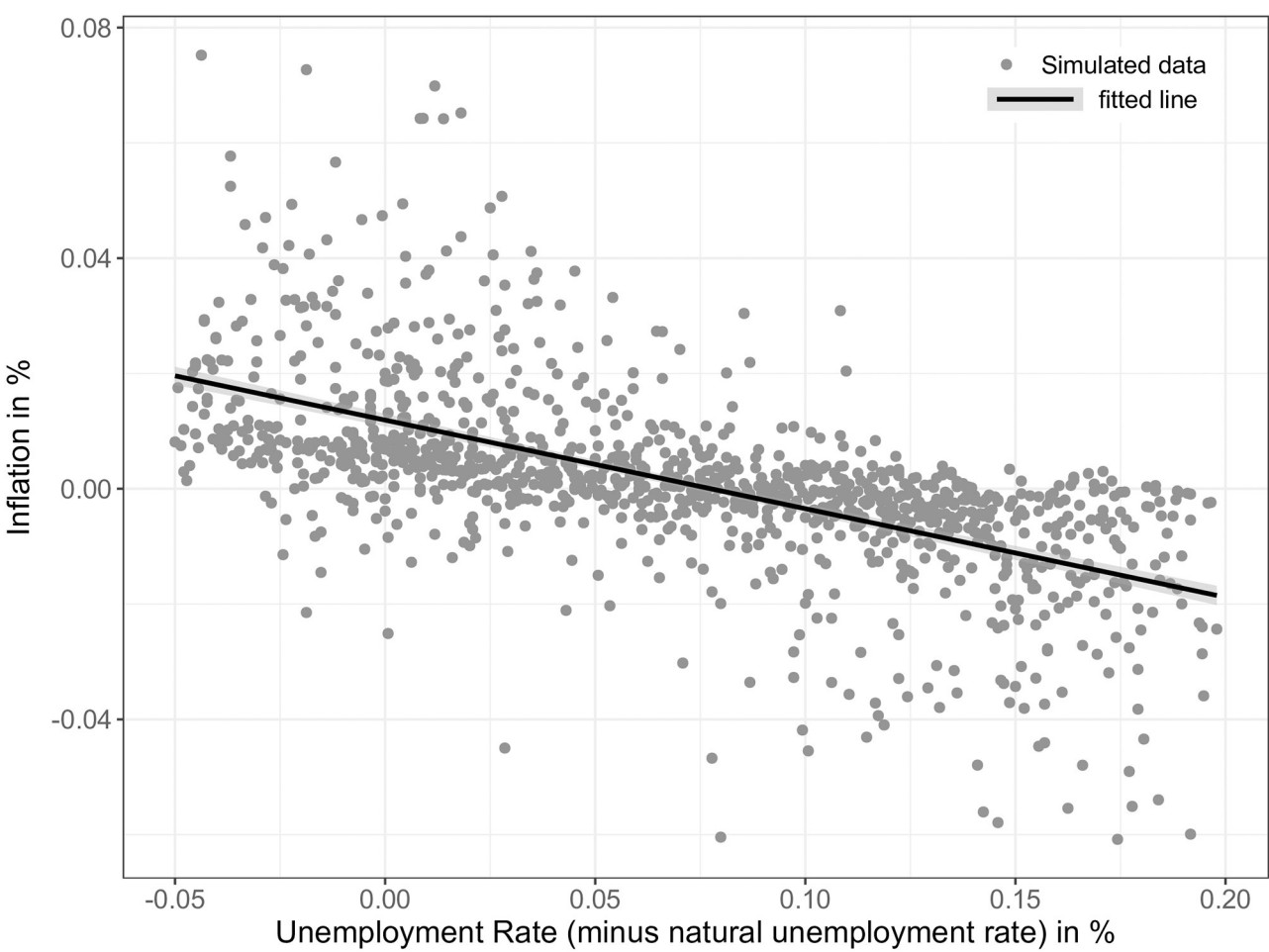

**Fig 12. SF11: Inflation in relation to unemployment. Note:** The Figure shows a scatter plot of average inflation percentage change and unemployment rate (minus natural unemployment rate) with following regression feature: *Natural unemployment rate* $(0.012 - 0.154x + \epsilon); R^2 = 0.369$. The natural unemployment rate is assumed to be 0.05. The simulation output comprises 20 representative runs to increase the sample with a total number of 12,000 period observations without the warm-up phase of 200 simulation periods. The grey shaded area represents the 90% confidence interval.

The credit cycle ( *requirement 2*) described in Section 3.5 is highly intertwined with the business cycle. Krug [29] investigates the interrelation between monetary and macroprudential policy, which requires a model with a focus on the cost-induced inflation theorem and omits a separate investment cycle. However, this approach does not capture credit overshooting behavior by banks as required. Excess demand by the purchasing power of credit supply is not incorporated since credit demand is restricted via the price-wage relation by firms and households. Mak(h)ro_0 simulates asset-price-related demand-induced inflation that omits a price-wage spiral. The corresponding rally of firms to reach the production potential makes the influence of credit supply on the business cycle more transparent, which is or can be covered by Caiani et al./Schasfoort et al. [28, 45] and van der Hoog [24]. Furthermore, Caiani et al./ Schasfoort et al. [28, 45], van der Hoog [24], and Krug [29] do not clarify how banks exploit and expand credit creation potential with quasi-automated fractional refinancing and circumvent capital adequacy and liquidity requirements. Concerning Popoyan et al. [30], the dynamic of the credit cycle is not transparent and accessible but seems to follow similar behavior as Krug [29].

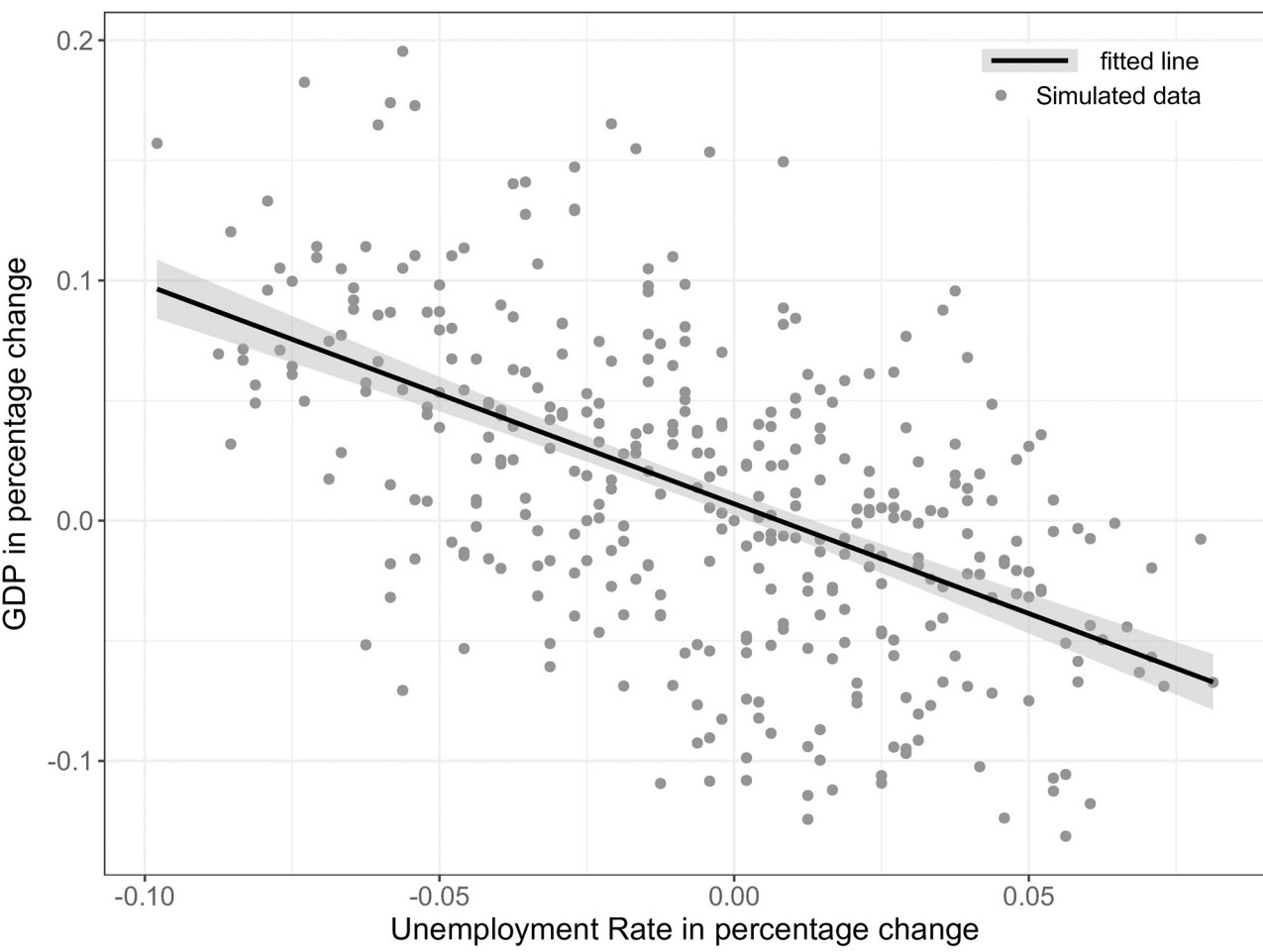

**Fig 13. SF12: GDP growth in relation to unemployment growth. Note:** The Figure shows a scatter plot of aggregated real GDP and unemployment percentage change with following regression features: *Unemployment rate* $(0.007 - 0.913x + \epsilon)$; $R^2 = 0.295$. The natural unemployment rate is assumed to be 0.05%. The simulation output comprises 20 representative runs to increase the sample with a total number of 12,000 period observations without the warm-up phase of 200 simulation periods. The grey shaded area represents the 90% confidence interval.

The fractional reserve system ( *requirement 3*) is highly interrelated with *requirement 2* and defines the institutional framework of the monetary system or how money creation is organized. Krug [29] exhibits a clear distinction between central bank money creation (reserves) and its relation to commercial bank money creation (two-split circuit system). This is in line with the theory of endogenous money creation (*requirement 4*) to identify liquidity risks of banks. Popoyan et al. [30] show an alternative way to implement the interbank market and to model the liquidity risk of banks while modeling bank liquidity management. However, a flow of reserves to fulfill the payment scheme, which is crucial for capturing the fractional reserve system, is not implemented. In comparison, even though Schasfoort et al. [28] included an interbank market, it is not clear how reserves are used for the payment scheme or how open market transactions are used to stabilize the interbank rate. Finally, van der Hoog [24] does not incorporate an interbank market and thus no adequate fractional reserve refinancing scheme to compare evolving liquidity risk across monetary regimes.

Endogenous money creation ( *requirement 4*) defines the money issuance process of reserves and credit money (Section 3.6.1). Whereas Caiani et al. [45] and Krug [29] strictly follow the endogenous money theory, van der Hoog [24] partly implements the *requirement 4* by

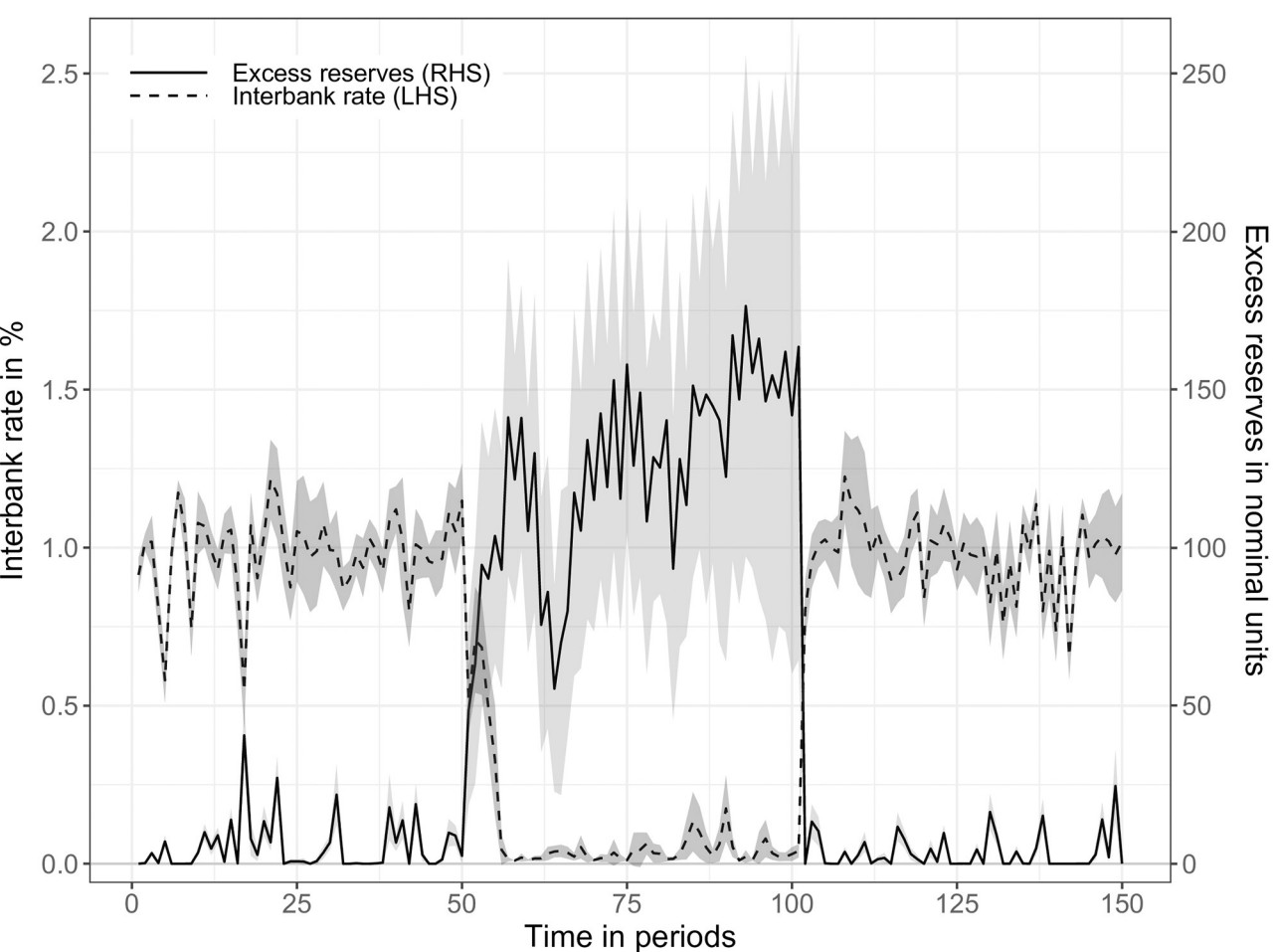

**Fig 14. SF13: Interbank market. Note:** The Figure depicts average interbank rate as a dashed line and excess reserves as a solid line. The simulation output is a representative run with an interbank experiment between period 50 and 100. The grey shaded area represents the 90% confidence interval.

**Table 5. Related literature in comparison to Mak(h)ro_0.**

| MABM | Model requirement | | | | | |
|---|---|---|---|---|---|---|
| | **1** | **2** | **3** | **4** | **5** | **6** |
| 1. Caiani/Schasfoort [28, 45] | ++ | ++ | + | ++ | + | ++ |
| 2. Van der Hoog [24] | ++ | ++ | − | + | + | ++ |
| 3. Krug [29] | ++ | + | ++ | ++ | ++ | + |
| 4. Popoyan [30] | ++ | + | + | − | + | ++ |

**Note:** Applicable (++); partly applicable (+); not applicable or identifiable (−)

The Table shows a comparison of related macroeconomic agent-based models (MABM) to the model at hand in accordance with requirements to model the current monetary system (as defined in Section 3.1).

Model requirement definition:

1. Business cycle; 2. Financial cycle; 3. Fractional reserve system; 4. Endogenous money creation; 5. Regulatory framework; 6. Monetary transmission

setting bank credit creation dependent on reserve requirements. However, reserve requirements play a minor role when it comes to credit restrictions within the current fractional reserve system [3, 41]. By assuming this restriction (as van der Hoog [24] does), the current monetary system might appear inherently more stable, which could bias the comparison with alternative monetary systems. Reserves, especially the minimum reserve, assumes the function of a liquidity buffer and are steered in the fractional reserve system via open market operations by the Central Bank (*requirement 3*). Finally, *requirement 4* can not be identified within the model description of Popoyan et al. [30].

**The regulatory framework ( *requirement 5*)** defines the macroprudential policy, which restricts the credit creation process of banks. Caiani et al./Schasfoort et al. [28, 45] and van der Hoog [24] do not implement the leverage ratio restriction for banks, which dampens banks' credit creation when leverage is high. In addition, Caiani et al./Schasfoort et al. [28, 45] and Popoyan et al. [30] do not specify an adaptation of risk-weighted assets according to the solvency state of firms through the financial cycle. This aspect could lead to banks overestimating their credit creation potential (see Eq 9), which impacts firms' demand via the purchasing power of credits [93].

Here we challenge the conclusion of van der Hoog [24] that only the quality of credit matters. In general, debtors and borrowers are confronted with asymmetric information which means that the quality of borrowers is not fully accessible. Therefore, the macroprudential policy according to Basel III (indirectly) addresses credit quantity to dampen excessive leverage. However, direct restrictions of banks' investment portfolios are more effective than indirect restrictions through capital, leverage, and liquidity regulations [94]. Furthermore, unproductive credits might also be caused by macroeconomic conditions and shocks. Credit rationing could worsen the financial situation for otherwise sound firms and might lead to contagion risks. Even when banks can choose unexceptionally productive credits or financially sound firms, systemic risk is still present.

**The monetary transmission mechanism ( *requirement 6*)** defines a basic monetary transmission mechanism that is necessary to analyze how monetary policy can control/influence the aggregate money supply in an economy. With respect to the model mechanism of Caiani et al./Schasfoort et al. [28, 45], van der Hoog [24] and Popoyan et al. [30] no significant differences can be detected. However, the model description of Krug [29] does not provide a financial accelerator mechanism which extends the credit creation potential of banks and thus increases the vulnerability of the economy.

The introduction provides a concise overview of related MABMs and the implementation of alternative monetary regimes. Beyond the indicated methodology, models incorporating a full reserve system are a Stock-Flow Consistent model [95], a sustainable-finance model based on differential equations [96], a sectoral accounting framework [97, 98], and an accounting system dynamics model [99], which can be classified as aggregate models. With this aggregated model approach, difficulties occur when analyzing heterogeneity within sectors, tracking intra-sectoral flows, and modeling asynchronous processes. The ability to model emergent crisis behavior as in MABMs is not achievable (see introduction and Section 3.1).

This Section underlines the statement in the introduction that there exists no MABM (nor DSGE model) specified to analyze the monetary system as summarized in Table 5. This research gap is closed with the combination of elaborating economic specifications for a monetary system (Section 3.1) and its implementation within a continuous-time stochastic agent-based simulation environment (ML3—Section 2.1 and provenance model—Section 2.2).

## 7 Conclusion

As stated in the introduction the aim of this model is to support studies for comparing alternative monetary regimes with a simple model of the current monetary system as a baseline. The model structure is based on pre-defined requirements (Section 3.1) and fully implemented within the continuous time agent-based modeling language ML3 (Section 2.1). ML3 allows to easily access, improve, expand, and discuss the model, not only to work with it but also to assess its quality. To analyze economic implications of monetary regime changes it is necessary that the model follows at least basic theoretical and/or empiric-economic interdependencies which have been made explicit within a provenance graph (Fig 3). It allows to easily identify and relate the various sources (such as theories or earlier simulation studies), separate mechanisms, submodels, and activities that contributed to generating Mak(h)ro_0, and it facilitates reuse and (together with the composite model design supported in ML3) future refinements and extensions. It can be shown that the model can replicate a wide range of stylized economic facts (Section 5) even with incisive limitations to reduce model complexity (Section 3.2.1). The replication of stylized facts is necessary to show that the simulated model macro-/microeconomic dynamics are plausible and capture characteristic patterns and distributions of empirical data. Hence, in the context of the defined model requirements, it can be concluded that the ABM "Mak(h)ro_0" presents a starting point for exploring and analyzing alternative monetary regimes in comparison with the current monetary system.

## Supporting information

**S1 Table. Entities and processes shown in Fig 3.** [18, 28, 29, 30, 42, 43, 45, 47, 50, 51, 54, 56, 63–65, 100].
(PDF)

**S2 Table. Balance sheet structure of all agent types.**
(PDF)

**S1 Fig. SF1: Deviations of real GDP and real consumption time series from trend: Band-pass-filtered (6,32,12). Note:** The Figure depicts deviations of average log real GDP (dashed line) and average log real household consumption (solid line) from trend. The simulation output is a representative run and shows an extract to keep the line profile more identifiable without the warm-up phase of 200 simulation periods. The grey shaded area represents the 90% confidence interval.
(TIF)

## Acknowledgments

We are grateful for helpful suggestions by all colleagues from the Department of Economics at the University of Rostock and the savings bank of Rostock for valuable practical insights. Furthermore, we acknowledge the Modeling and Simulation Group of the Institute of Visual and Analytic Computing at the University of Rostock for valuable criticisms and suggestions that helped improve this paper substantially.

## Author Contributions

**Conceptualization:** Florian Peters, Doris Neuberger.

**Data curation:** Florian Peters.

**Formal analysis:** Florian Peters.

**Funding acquisition:** Adelinde Uhrmacher.

**Investigation:** Florian Peters.

**Methodology:** Florian Peters, Oliver Reinhardt.

**Project administration:** Florian Peters.

**Resources:** Florian Peters, Doris Neuberger, Adelinde Uhrmacher.

**Software:** Florian Peters, Oliver Reinhardt.

**Supervision:** Doris Neuberger, Adelinde Uhrmacher.

**Validation:** Florian Peters.

**Visualization:** Florian Peters.

**Writing – original draft:** Florian Peters, Oliver Reinhardt.

**Writing – review & editing:** Florian Peters, Doris Neuberger, Oliver Reinhardt, Adelinde Uhrmacher.

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
