## [Decision Letter · Decision Letter 0]

5 Jul 2022

PONE-D-22-13055

A basic macroeconomic agent-based model for analyzing monetary regime shifts

PLOS ONE

Dear Dr. Peters,

Thank you for submitting your manuscript to PLOS ONE. After careful consideration, we feel that it has merit but does not fully meet PLOS ONE’s publication criteria as it currently stands. Therefore, we invite you to submit a revised version of the manuscript that addresses the points raised during the review process.

From their reviews, you will see that our two referees consider your paper, interesting, novel and with potential great interest to the academical community, so do I. One of them has accepted your article giving you some future suggestion on your methodology. The other reviewer expresses some concerns and questions on your main contribution and ask you to clarify some ponts related to your exposition and conclusions.   Please answer all comments and worries of our referees and submit your revised manuscript by Aug 19 2022 11:59PM. If you will need more time than this to complete your revisions, please reply to this message or contact the journal office at plosone@plos.org. Please include the following items when submitting your revised manuscript:A rebuttal letter that responds to each point raised by the academic editor and reviewer(s). You should upload this letter as a separate file labeled 'Response to Reviewers'.A marked-up copy of your manuscript that highlights changes made to the original version. You should upload this as a separate file labeled 'Revised Manuscript with Track Changes'.An unmarked version of your revised paper without tracked changes. You should upload this as a separate file labeled 'Manuscript'.If applicable, we recommend that you deposit your laboratory protocols in protocols.io to enhance the reproducibility of your results. Protocols.io assigns your protocol its own identifier (DOI) so that it can be cited independently in the future. For instructions see: https://journals.plos.org/plosone/s/submission-guidelines#loc-laboratory-protocols. Additionally, PLOS ONE offers an option for publishing peer-reviewed Lab Protocol articles, which describe protocols hosted on protocols.io. Read more information on sharing protocols at https://plos.org/protocols?utm_medium=editorial-email&utm_source=authorletters&utm_campaign=protocols.

We look forward to receiving your revised manuscript.

Kind regards,

Alejandro Raúl Hernández-Montoya, Ph D

Academic Editor

PLOS ONE

Journal Requirements:

2. Please ensure that you refer to Figures 6,10,12 and 15 and in your text as, if accepted, production will need this reference to link the reader to the figure.

Reviewers' comments:

Reviewer's Responses to Questions

**Comments to the Author**

1. Is the manuscript technically sound, and do the data support the conclusions?

Reviewer #1: Yes

Reviewer #2: Yes

2. Has the statistical analysis been performed appropriately and rigorously? 

Reviewer #1: Yes

Reviewer #2: Yes

3. Have the authors made all data underlying the findings in their manuscript fully available?

Reviewer #1: Yes

Reviewer #2: Yes

4. Is the manuscript presented in an intelligible fashion and written in standard English?

Reviewer #1: Yes

Reviewer #2: Yes

5. Review Comments to the Author

Reviewer #1: I am somewhat sympathetic towards the basic idea and approach of the paper. The study is concise, relatively easy to follow, well written and the empirical work appears to be correctly done. The study addresses an interesting and very timely topic which is of relevance for policymakers. Although I find the topic is interesting and appreciate the effort of the authors in testing the robustness of their results, at the same time, I consider that the setup is not clear. My main concern with the paper is its incremental contribution. I also think that the paper fails to deliver any general economic insights. Nonetheless, a revision is required to address several concerns.

a) My biggest concern with the paper is its contribution. How exactly does your paper contribute to the debate on the alternative monetary regimes? Your paper does not make it clear to me or to potential readers of the Journal what your unique contribution to the literature is.

b) The introduction should build a succinct motivation for the paper and describe how the paper extends the prior literature. The author has to clarify from the beginning the importance of this kind of exercise and the differences with the researcher already done in the literature.

c) The authors should put more emphasis on discussing and interpreting their findings. The authors should discuss more the results and not just describe it. The paper lacks economic intuition and fails to deliver any general economic insights. What are the economic consequences of the results? In my opinion, the authors fail to answer the "so what?" question.

Reviewer #2: I enjoyed this paper, and I am curious to see the academic community's reaction. Here are the following reasons for submitting my recommendation as "accept":

The simulation is well thought out, and thorough in its attempt to the simulate the agents involved.

The applications are very real.

The environment is complex, yet it has been approached with scientific rigor, and I appreciate that.

My suggestions:

The authors acknowledge the complexities of the problem in the Introduction. They also acknowledge the nonlinearties that exist in their data. This is all setup to their choosing a MABM models over DSGE models. I also reviewed the graph of their environment in their appendix and conclude that it is a fairly complicated graph.

Due to nonlinearities in their data and graph like nature of their problem, I would suggest future simulations would benefit from artificial intelligent techniques, including reinforcement learning agents. These agents could easily handle the nonlinearities that exist in the data, and would be more adaptive than traditional models as the environment becomes more complex.

In anticipation of the authors response to my suggestion that the models they have selective allow them to be deterministic and interpretable in their results- I would suggest more recent machine learning models are also deterministic and interpretable. For example XGBoost.

Overall, a very good paper.

6. PLOS authors have the option to publish the peer review history of their article (what does this mean?). If published, this will include your full peer review and any attached files.

Reviewer #1: No

Reviewer #2: No

---

## [Author Response · Author response to Decision Letter 0]

29 Aug 2022

1 General Remarks: Response to Reviewers

• First of all we would like to thank the editor and the reviewers for their detailed comments to improve the paper.

• We have rewritten the introduction, to make the focus and the contribution more clear.

• We included more information in the introduction, to emphasize the research gap.

• We adjusted the Abstract and Section 6 (Discussion and literature review) in accordance to the introduction.

2 Editors’ Comments to Authors

From their reviews, you will see that our two referees consider your paper, interesting, novel and with potential great interest to the academical community, so do I. One of them has accepted your article giving you some future suggestion on your methodology. The other reviewer expresses some concerns and questions on your main contribution and ask you to clarify some ponts related to your exposition and conclusions.

3 Reviewers’ Comments to Authors

3.1 Referee: 1

3.1.1 Comments to Authors

I am somewhat sympathetic towards the basic idea and approach of the paper. The study is concise, relatively easy to follow, well written and the empirical work appears to be correctly done. The study addresses an interesting and very timely topic which is of relevance for policymakers. Although I find the topic is interesting and appreciate the effort of the authors in testing the robustness of their results, at the same time, I consider that the setup is not clear. My main concern with the paper is its incremental contribution. I also think that the paper fails to deliver any general economic insights. Nonetheless, a revision is required to address several concerns.

a) My biggest concern with the paper is its contribution. How exactly does your paper contribute to the debate on the alternative monetary regimes? Your paper does not make it clear to me or to potential readers of the Journal what your unique contribution to the literature is.

• To highlight the contribution of the paper in the introduction, we first included more information on the definition of the monetary system and its importance to the financial system (lines 8 to 10): It is the underlying “software” of the financial system that enables payments, savings, insurance, and financing. And second, we highlight the main issue of dissent in the current academic debate (lines 13 to 17): The main issue of dissent is whether the current system is inefficient and ineffective in terms of macro-financial stability compared with available alternatives (e.g., optimal money supply, effective financial intermediation, shadow banking).

• We are grateful for the hint to clarify the purpose of the paper, which focuses on the methodological approach rather than on the implementation of alternative monetary systems and corresponding economic implications. We included more information (lines 25 to 28) and adjusted the abstract.

• Section 6 (Discussion and literature review) states the concrete contribution to the literature. However, we totally agree that the introduction should elucidate the research gap in a succinct and conclusive manner which also includes literature or models beyond the agent-based approach. Since this concern relates to point b) we continue our explanations in the next point.

b) The introduction should build a succinct motivation for the paper and describe how the paper extends the prior literature. The author has to clarify from the beginning the importance of this kind of exercise and the differences with the researcher already done in the literature.

• As stated in point a), we agree with this observation. The extension to the literature has been summarized in the introduction (lines 57 to 105) and is structured in the following paragraphs: First, the related literature with respect to the debate of monetary reforms has been set in relation to our research. This underlines the research gap in a more detailed manner and shows the contextual intricacies of monetary reforms (lines 57 to 78). Second, we highlight the twofold contribution of our model to the literature: methodological (lines 79 to 89) and economic (lines 90 to 105) contribution. The methodological contribution involves the model implementation (MABM) within a continuous-time stochastic agent-based simulation environment and the corresponding provenance model. On the other hand, the economic contribution consists in defining the model requirements and the related economic model.

• We have emphasized the conclusion of the paper in lines 106 to 109, which further reinforces the purpose of the paper.

• In the course of describing the research gap in more detail in the introduction, the paragraph to compare related literature beyond the MABM/DSGE approach in Section 6 was adjusted (lines 1027 to 1035). Here we added a sectoral accounting framework from Flaschel et al. (2010) and Chiarella (2012).

c) The authors should put more emphasis on discussing and interpreting their findings. The authors should discuss more the results and not just describe it. The paper lacks economic intuition and fails to deliver any general economic insights. What are the economic consequences of the results? In my opinion, the authors fail to answer the ”so what?” question.

• This a valid point. We agree that potential readers could misunderstand the purpose of the paper when expecting the implementation of monetary systems and its economic implications. Thanks to concerns a) and b) and the corresponding modifications as described above, we now see little potential for misunderstandings. We aim for describing the model and its settings rather than proceed economic experiments (see also point a)). This procedure has been chosen due to the complexity of the research project and the scale of the model. We see the description of the model and its economic setting as an important task to let other researchers understand, replicate and apply the model. The additional execution and description of economic experiments (e.g., the implementation of alternative monetary systems) would exceed the scope of this research paper. Next to the conclusion, we also clarified the answer to the “so what?” question in the abstract and introduction: “[...] presents a starting point for exploring and analyzing monetary reforms [...]”

• Furthermore, we see the answer to the “so what?” question in the combination of elaborating economic specifications for a monetary system and its implementation within a novel simulation environment (ML3) which defines the contribution to the MABM literature. We can conclude that the literature does not reveal a MABM that analyzes a monetary system. This research gap has been closed with our contribution. We added a corresponding paragraph at end of Section 6.

• Additionally, we see that Table 5 might be misleading in the sense that is not clear the related MABMs are compared with the model at hand. Therefore, the table description and the first paragraph of Section 6 has been adjusted.

• The result section represents a general validation outcome based on economic stylized facts. This validation technique is commonly used within the agent-based community and is based on empiric observations as shown in Table 1. Here we see little potential for a discussion, since we present standardized techniques that are used to analyze macroeconomic behavior.

3.2 Referee: 2

3.2.1 Comments to Authors

I enjoyed this paper, and I am curious to see the academic community’s reaction.

The simulation is well thought out, and thorough in its attempt to the simulate the agents involved. The applications are very real. The environment is complex, yet it has been approached with scientific rigor, and I appreciate that. The authors acknowledge the complexities of the problem in the Introduction. They also acknowledge the nonlinearties that exist in their data. This is all setup to their choosing a MABM models over DSGE models. I also reviewed the graph of their environment in their appendix and conclude that it is a fairly complicated graph.

Due to nonlinearities in their data and graph like nature of their problem, I would suggest future simulations would benefit from artificial intelligent techniques, including reinforcement learning agents. These agents could easily handle the nonlinearities that exist in the data, and would be more adaptive than traditional models as the environment becomes more complex.

In anticipation of the authors response to my suggestion that the models they have selective allow them to be deterministic and interpretable in their results- I would suggest more recent machine learning models are also deterministic and interpretable. For example XGBoost.

• We appreciate the suggestions. We do indeed see transparency and interpretability as important factors (less so determinism, and our model is indeed stochastic - see the outline of the framework in Section 2). We can imagine to replace some of the decision-making strategies of the agents with artificial intelligence techniques, e.g., do firms use a reinforcement-learning strategy to determine prices. It might be interesting to see if such strategies could yield a more realistic pricing behavior than the existing simple rules. The component-based construction of our model would allow such a replacement. However, in our validation we found that the current model does already reproduce the desired behavior sufficiently accurately - and there is something to be said in favour of a simple model, if it is sufficiently accurate for it’s purpose.

Overall, a very good paper.

• Thank You!

---

## [Decision Letter · Decision Letter 1]

1 Nov 2022

A basic macroeconomic agent-based model for analyzing monetary regime shifts

PONE-D-22-13055R1

Dear Dr. Peters,

We’re pleased to inform you that your manuscript has been judged scientifically suitable for publication and will be formally accepted for publication once it meets all outstanding technical requirements.

Kind regards,

Rodrigo Huerta-Quintanilla, Ph. D

Academic Editor

PLOS ONE

Additional Editor Comments (optional):

Reviewers' comments:

Reviewer's Responses to Questions

**Comments to the Author**

1. If the authors have adequately addressed your comments raised in a previous round of review and you feel that this manuscript is now acceptable for publication, you may indicate that here to bypass the “Comments to the Author” section, enter your conflict of interest statement in the “Confidential to Editor” section, and submit your "Accept" recommendation.

Reviewer #1: All comments have been addressed

Reviewer #2: All comments have been addressed

2. Is the manuscript technically sound, and do the data support the conclusions?

Reviewer #1: Yes

Reviewer #2: Yes

3. Has the statistical analysis been performed appropriately and rigorously? 

Reviewer #1: Yes

Reviewer #2: Yes

4. Have the authors made all data underlying the findings in their manuscript fully available?

Reviewer #1: Yes

Reviewer #2: Yes

5. Is the manuscript presented in an intelligible fashion and written in standard English?

Reviewer #1: Yes

Reviewer #2: No

6. Review Comments to the Author

Reviewer #1: The authors have put a lot effort to improve the paper and reply to all my comments/recommendations.

Reviewer #2: I appreciated their comments back to me. While it did not require any immediate edits to the current paper, I like how they are thinking about the problem in the future.

7. PLOS authors have the option to publish the peer review history of their article (what does this mean?). If published, this will include your full peer review and any attached files.

Reviewer #1: No

Reviewer #2: **Yes: **Andrew Brim

---

## [Editor Report · Acceptance letter]

14 Nov 2022

PONE-D-22-13055R1 

A basic macroeconomic agent-based model for analyzing monetary regime shifts 

Dear Dr. Peters:

I'm pleased to inform you that your manuscript has been deemed suitable for publication in PLOS ONE. Congratulations! Your manuscript is now with our production department. 

Kind regards, 

on behalf of

Dr. Rodrigo Huerta-Quintanilla 

Academic Editor

PLOS ONE